# Ensemble intelligence prediction algorithms and land use scenarios to measure carbon emissions of the Yangtze River Delta: A machine learning model based on Long Short-Term Memory

**Qi Dai**[1‡], **Xiao-yan Liu**[2‡], **Fang-yi Sun**[2], **Fang-rong Ren**[2]*

**1** College of Public Administration, Hohai University, Nanjing, P.R. China, **2** College of Economics and Management, Nanjing Forestry University, Nanjing, P.R. China

‡ QD and XL have contributed equally to this work and share first authorship on this work.
* 180213120008@hhu.edu.cn

**Data Availability Statement:** All relevant data are within the manuscript and its Supporting information files.

## Abstract

Land use in urban agglomerations is the main source of carbon emissions, and reducing them and improving land use efficiency are the keys to achieving sustainable development goals (SDGs). To advance the literature on densely populated cities and highly commercialized regions, this research evaluates the total-factor carbon emission efficiency index (TCEI) of 27 cities in China's Yangtze River Delta (YRD) urban agglomeration at different stages from 2011 to 2020 using two-stage dynamic data envelopment analysis (DEA). The study carries out regression analysis and a long-short-term memory model (LSTM) to respectively filter out the factors and predict TCEI. The results indicate the following. (1) The total efficiency of 27 cities has significantly improved from 2011 to 2020, and there are obvious spatial heterogeneity characteristics. (2) In terms of stages, most cities' efficiency values in the initial stage (energy consumption) exceed those in the second stage (sustainable land utilization). (3) In terms of influencing factors, urban green space's ability to capture carbon has a notably positive correlation with carbon emission efficiency. In contrast, the substantial carbon emissions resulting from human respiration are a negative factor affecting carbon emission efficiency. (4) Over the forthcoming six years, the efficiency value of land use TCEI in the YRD urban cluster is forecasted to range between 0.65 and 0.75. Those cities with the highest performance are projected to achieve an efficiency value of 0.9480. Lastly, this research investigates the interaction between actors and land resources on TCEI, resulting in a beneficial understanding for the former to make strategic adjustments during the urbanization process.

## 1 Introduction

Carbon emissions have become a major concern worldwide ever since the industrial revolution. Currently, carbon dioxide is the main greenhouse gas (GHG) produced by human

**Funding:** This study was supported by Jiangsu Province Social Science Foundation Project (22GLD019), Major Project of Philosophy and Social Science Research in Universities of Jiangsu Province(2022SJZD053).

**Competing interests:** The authors have declared that no competing interests exist.

**Abbreviations:** AVE, Average; CNY, Chinese yuan; DEA, Data envelopment analysis; DMU, Decision-making unit; IPCC, Intergovernmental Panel on Climate Change; LMDI, Logarithmic mean divisia index; LSTM, Long Short-Term Memory; MAE, Mean absolute error; MPI, Malmquist productivity index; MSE, Mean-square error; RNN, Recurrent neural network; SBM, Slacks-based measure; SFA, Stochastic frontier model; St. dev., Standard deviation; TCEI, Total-factor carbon emission efficiency index; YRD, Yangtze River Delta.

activities. The Intergovernmental Panel on Climate Change (IPCC) has indicated that worldwide temperatures could rise by as much as 1.5 Celsius as soon as 2030, and that the world will likely face unavoidable multiple climate hazards in the next two decades. China's carbon emissions in 2007 exceeded those of the United States and other developed nations, making it the world's most carbon-intensive country. In 2010, its energy consumption reached the highest level in the world. Research has suggested [1] that China's carbon emissions in 2030 will increase by 30% versus 2010.

With the aim of managing rising temperatures, China has undertaken numerous initiatives in the area of carbon reduction. The country has set forth a strategy to reach peak carbon emissions by 2030 and achieve carbon neutrality by 2060. The sustainable development agenda of the United Nations forecasts by 2030 that the rate of global urbanization will hit approximately 60%. At the same time, urban areas contribute about 70% of carbon emissions, and this proportion is expected to rise by about 6% in a decade. With China's urbanization rate already at 66.16% in 2023, this rapid expansion into non-built-up land has become an inevitable trend. Accelerated urbanization not only leads to greater carbon intensity and energy consumption, but also poses a significant risk to both the ecological environment and human living conditions.

Land use contributes significantly to GHG emissions. Indeed, the process of land use generates 23% of total global carbon emissions, while at the same time it absorbs carbon dioxide equivalent to about 33% of worldwide carbon emissions from industrial production and fossil fuel combustion. The YRD city cluster is the most economically efficient and urban agglomerated city area in China, accounting for 20% of domestic economic output despite occupying only 2.1% of its land area. The total permanent population of YRD in 2023 was 236.9 million people, reflecting the population agglomeration effect of this important region. Considering the lack of case studies specifically aimed at densely populated cities and highly commercialized areas and the need for empirical application of demonstrative methods, this paper chooses YRD as a case to evaluate the path of energy conservation and emission reduction of urban agglomerations with good economic endowment in developing countries [2, 3].

This research takes YRD as the sample target to first measure and analyze its carbon emission efficiency through a two-stage dynamic DEA model so as to clarify the trend of TCEI changes among cities in the city cluster. Moreover, it employs the Tobit model to pinpoint the factors driving carbon emission efficiency in the YRD city cluster. Finally, an LSTM neural network model is utilized to forecast changes in TCEI.

This paper's innovations and major contributions cover four main points. First, unlike studies that measured carbon emission efficiency through a single dimension of energy consumption, this study takes land use in urban agglomerations as a new entry point, and further optimize the urban carbon emission efficiency index system. Second, the dynamic two-stage DEA model resolves the issue of subjective assignment of specific production functions in the traditional DEA model and broadens the notion of TCEI measurement. Third, to identify influencing factors, this study quantifies the direction and degree of influence of industrial structure, urban population density, and urban environmental governance level on TCEI, in order to help the government to identify pollution sources and to provide a basis for exploring the sustainable use of land in urban agglomerations and achieving the dual-carbon goal. Fourth, at the level of empirical methods, this study combines DEA with the neural network model from the field of artificial intelligence (AI), thus expanding the modeling approaches taken for predicting carbon emission efficiency. It offers a certain degree of generalizability.

## 2 Literature review

### 2.1 Conceptual definition of carbon emissions from land use

First, since land is an important carbon sink and source in the ecosystem, assessing carbon emission efficiency in land use is crucial for evaluating the effectiveness of the sustainable development model. Other studies on land use carbon emission efficiency mainly have focused on evaluating such efficiency under different types of land use and various multi-dimensional indicators.

Second, carbon emission efficiency is commonly divided by most researchers into two categories: single-factor carbon emission efficiency and multi-factor carbon emission efficiency. Single-factor carbon emission efficiency is measured by the GDP created per unit of carbon emission [4], whereas total factor efficiency is usually associated with socio-economic indicators, reflecting more precisely the comprehensive efficiency of carbon emissions compared to a single-factor viewpoint [5–7]. Therefore, the current study treats carbon dioxide emission as a non-desired output to measure TCEI in the land use process of YRD urban agglomeration.

### 2.2 Factors affecting carbon efficiency of land use

Identifying the influencing factors is an important prerequisite before predicting TCEI. The literature underscores a variety of elements influencing carbon emission efficiency, which include the structure of energy, industrial structure, and the socio-economic level. Most studies have adopted methods such as environmental Kuznets curve, LMDI model, geographic detector, Tobit model, and so on.

Cointegration analysis was initially used by some scholars [8] to identify the factors affecting carbon emissions in China's textile industry. Some studies [9, 10] later proposed an extended exponential decomposition method through the STIRPAT model. Other scholars [11] combined the logarithmic mean divisional index (LMDI) method and two-stage DEA to measure an economy's environmental efficiency and found that population migration and mobility, labor force scheme, and energy consumption intensity are important affecting factors of energy consumption. Nevertheless, the development of economic models, like LMDI and STIRPAT, frequently requires the application of an assortment of assumptions and is subject to limitations. The characteristics of data volatility are not sufficiently explained by econometric models.

Tobin (1958) introduced the Tobit regression model, which is a type of limited dependent variable regression technique. This model depicts the relationship between a non-negative dependent variable (also referred to as a latent variable) and an independent variable in circumstances where data are either censored or truncated. However, a more in-depth exploration of the macro-drivers of logistics efficiency can be achieved through the dynamic two-stage DEA model, which combines DEA and Tobit regression. This model offers scientific references for policy making aimed at enhancing land use efficiency and realizing sustainable development. For example, Liu [12] applied the Tobit econometric model to assess the influence on logistics efficiency following implementation of the SBM-DEA method. This suggests that the integration of DEA and the Tobit model is reliable for depicting efficiency in the logistics industry. However, most studies did not forecast the efficiency value after identifying the factors influencing TCEI. To address this gap in the literature, this study employs the identified affecting factors as input variables in a forecasting model to evaluate their predictive bias and drivers.

### 2.3 Application of DEA methodology to land use efficiency assessment

First, when measuring the efficiency of land use carbon emissions, most studies have used stochastic frontier analysis (SFA) and DEA [13, 14] for relative efficiency analysis. SFA often

requires the establishment of an ideal f-value, and this excessively stringent assumption may result in structural prejudice as a result of an incorrect manufacturing procedure setting [15, 16]. Conversely, DEA is widely employed in the field of carbon emission performance, because it directly calculates data without the necessity of pre-estimating parameters [17]. The call for urban development to reduce energy consumption and associated carbon emissions has become more prevalent over the past decade as a result of the pressure to return to green and sustainable development. Furthermore, the conventional DEA model is frequently less precise than the two-stage DEA model with carry-over variables. As a result, many scholars have included carbon emission indicators as undesirable outputs in efficiency assessments [18–20].

Reinhard [21] and Hailu [22] defined pollution variables as input factors and applied the DEA model for ecological performance evaluation under the full-factor structure, but it has some limitations due to the deviation between the model and the real manufacturing procedure. Based on this, some studies [7] considered carbon emissions as undesirable outputs, which makes the results of the model considering undesirable output variables more accurate and reliable. However, since all the above models assumed that each DMU has the same production boundary, which is obviously inconsistent with the actual situation, the conventional DEA model cannot unfold the technological differences and growth patterns of each DMU in the land use process [23–25]. Some studies noted [26] that network DEA models after adding slack variables effectively analyze the efficiency of individual sectors and also uncover the correlations between stages. Therefore, a weighted SBM dynamic network DEA model has been proposed for calculating multi-stage efficiency values. The two-stage dynamic DEA model, which this research refers to, considers the influence of undesirable outputs on the efficiency value and also uses the interconnection among the DMU departments as the foundation for efficiency measurement.

This research thus adopts the dynamic two-stage DEA modelling approach in carrying out measurement of land use TCEI. It integrates the correlation of input and output variables in the two stages of urban agglomeration's land use TCEI and analyzes the independent effect and connecting effect of S1 (energy consumption stage) and S2 (sustainable land use stage), respectively. This makes it a more unique and segmented research approach.

In predicting future carbon emission efficiency, most studies have utilized methods such as the STIRPAT model, Leap model, grey correlation model, system dynamics model, etc. Xu et al. predicted that carbon emissions will peak in China between 2029–2035 through the STIRPAT model [27]. Ding et al. proposed that without a carbon tax, China is expected to hit its peak carbon in 2030 [28]. Nonetheless, leveraging the diverse carbon emission efficiency correlation data is challenging due to conventional statistical modeling and prediction methods. Hence, AI algorithms can address data analysis and prediction issues in the energy and environment sectors and have emerged as a new area of research interest [29].

Long Short-Term Memory (LSTM), which is also known as Long Short-Term Memory structure, is a variant of traditional RNN and can efficiently predict the long- and short-term dynamic trends of time series data. Ouyang [30] compared the LSTM model with four models, including GARCH, and empirically found that LSTM has higher forecasting accuracy [31] and captures the semantic correlation between long time series. Most research centers around measuring carbon emission efficiency in urban land use processes and analyzing the factors that drive it [32–35]. There are fewer studies combining the measurement of the efficiency value with the neural network model, and an extended prediction of future efficiency values for urban agglomerations is lacking. Based on this, Athanassopoulos and Curram [36] proposed the idea of combining DEA with machine learning and used DEA for training set screening and preprocessing and ANN tool for prediction of non-linear models in the field of carbon emissions.

In summary, there are some shortcomings that arise in other studies' results. First, the majority of them has focused on the accounting for TCEI of specific land use types, thereby overlooking macro-level land use efficiency measurement. This also results in a decline in the number of analyses concerning the driving factors of a region's overall land use TCEI. Second, many studies have ignored the effects of non-expected variables and carry-over variables in the process of calculating TCEI value by DEA, which means that the calculation results do not objectively reflect the dynamic characteristics of land use carbon emissions. Third, the dynamic two-stage DEA model is able to assess the whole-factor carbon emission efficiency of different land resource allocation levels at the level of urban agglomerations through a more scientific manner. Ultimately, most studies have primarily concentrated on determining TCEI of land use and lacked a comprehensive forecast of future efficiency values for urban conglomerations. Therefore, this paper presents the two-stage dynamic DEA-Tobit-LSTM model in the realm of AI that can efficiently interpret the forecast results by contrasting the outcomes of dynamic scenario simulation using the deep learning approach. This supplements the body of literature related to the prediction of trends in land use carbon emissions.

## 3 Research method

### 3.1 Dynamic two-stage DEA approach

This study addresses issues arising from static analyses and regional disparities by segmenting the approach into two stages: the first stage (S1) is energy consumption, and the second stage (S2) is sustainable land use. Assume that there are $n$ *DMUs* ($j = 1,\ldots, n$), and the *DMUs* are 27 cities of YRD in this research. Each one has $k$ partitions ($k = 1,\ldots, k$) and $n$ time periods ($T = 1,\ldots, T$), and the time series is 2011–2020 in this paper. Each *DMU* has an input and an output in time period $T$ and a carry-over (link) in the next time period $T + 1$ to the next time period ($j = 1,\ldots, n$). The inputs and outputs of each partition $K$ by $K$ ($K, h$)$i$ denote the divisions from $k$ to $h$, and $n$ and $Lhk$ denote the set of divisions from $k$ and $h$. We show the inputs and outputs, links, and carry-over definitions as follows.

Inputs and outputs:

$X_{ijk}^t \in R_+ (i = 1,\ldots, m_k; \rightleftarrows j = 1,\ldots, n; K = 1\ldots, K; t = 1,\ldots, T)$: refers to input $i$ at time period $t$ for $DMU_j$ division $k$.

$y_{rjk}^t \in R_+ (r = 1,\ldots, r_k; \rightleftarrows j = 1,\ldots, n; K = 1\ldots, K; t = 1,\ldots, T)$: refers to output $r$ in time period $t$ for $DMU_j$ division $k$.

Links:

$Z_{j(kh)t}^t \in R_+ (j = 1; \ldots; n; l = 1; \ldots; L_{hk}; t = 1; \ldots; T)$: refers to the period $t$ links from $DMU_j$ (b) the process of time during which the scheme is to be implemented. The following is a conclude of the period of the scheme of work $h$. Here, $L_{hk}$ represents the number of $k$ links from one division to another, while $h$ links refer to the quantity.

$Z_{j(kh)t}^t \in R_+ (j = 1; \ldots; n; l = 1; \ldots; L_{kh}; t = 1; \ldots; T)$.

Carry-overs:

$Z_{jkl}^{(t,t+1)} \in R_+ (j = 1,\ldots, n; l = 1,.., L_k; k = 1,\ldots k, t = 1,\ldots, T-1)$: refers to the carry-over from time period (t) to period (t+1). This includes division $h$ to division $k$, where $L_k$ represents the number of carry-over items in division $k$. The quantity of input links for every division $k$ is represented by Link$in_k$. Similarly, Link $out_k$ signifies the quantity of output links for every division $k$. Each division $k$ has a certain number of favorable carry-overs, represented by $n\ good_k$. Conversely, the number of unfavorable carry-overs for each division is represented by the quantity of output links for each division, denoted by $n\ bad_k$.

What follows is the model that is non-oriented.

(a) Goal function

Overall efficiency.

$$\theta_0^* = \min \frac{\sum_{t=1}^{T} W^t \left[ \sum_{k=1}^{K} W^k \left[ 1 - \frac{1}{m_k + linkin_k + input_k} \left( \sum_{i=1}^{m_k} \frac{S_{iok}^{t-}}{x_{iok}^t} + \sum_{(kh)_l=1}^{linkin_k} \frac{s_{o(kh)_lin}^t}{z_{o(kh)_lin}^t} + \sum_{k_l}^{ninput_k} \frac{s_{ok_linput}^{(t,t+1)}}{z_{ok_linput}^{(t,t+1)}} \right) \right] \right]}{\sum_{t=1}^{T} W^t \left[ \sum_{k=1}^{K} W^k \left[ 1 + \frac{1}{r_{1k} + r_{2k}} \left( \sum_{r=1}^{r_{1k}} \frac{s_{rokgood}^{t+}}{y_{rokgood}^t} + \sum_{r=1}^{r_{2k}} \frac{s_{rokbad}^{t-}}{y_{rokbad}^t} \right) \right] \right]} \tag{1}$$

Subject to.

$$x_{ok}^t = X_k^t \lambda_k^t + s_{ko}^{t-} (\forall k, \forall t) \tag{2}$$

$$y_{okgood}^t = Y_{kgood}^t \lambda_k^t - s_{kogood}^{t+} (\forall k, \forall t) \tag{3}$$

$$y_{okbad}^t = Y_{kbad}^t \lambda_k^t + s_{kobad}^{t-} (\forall k, \forall t) \tag{4}$$

$$e \lambda_k^t = 1 (\forall k, \forall t)$$
$$\lambda_k^t \geq 0, s_{ko}^{t-} \geq 0, s_{kogood}^{t+} \geq 0, s_{kobad}^{t-} \geq 0, (\forall k, \forall t) \tag{5}$$

$$Z_{o(kh)in}^t = Z_{(kh)in}^t \lambda_k^t + S_{o(kh)in}^t ((kh)in = 1, \ldots, linkin_k) \tag{6}$$

$$\sum_{j=1}^{n} z_{jk_1\alpha}^{(t,(t+1))} \lambda_{jk}^t = \sum_{j=1}^{n} z_{jk_1\alpha}^{(t,(t+1))} \lambda_{jk}^{t+1} (\forall k; \forall k_l; t = 1, \ldots, T-1) Z_{ok_linput}^{(t,(t+1))}$$

$$= \sum_{j=1}^{n} z_{jk_linput}^{(t,(t+1))} \lambda_{jk}^t + s_{ok_linput}^{(t,(t+1))} k_l = 1, \ldots, ngood_k; \forall k; \forall t \right) \tag{7}$$

$$s_{ok_lgood}^{(t,(t+1))} \geq 0, (\forall k_l; \forall t)$$

(b) Efficiencies of period and division

Efficiencies of period and division are as follows.

i. Period efficiency.

$$\partial_0^* = \min \frac{\sum_{k=1}^{K} W^k \left[ 1 - \frac{1}{m_k + linkin_k} \left( \sum_{i=1}^{m_k} \frac{S_{iok}^{t-}}{x_{iok}^t} + \sum_{(kh)_l=1}^{linkin_k=1} \frac{s_{o(kh)_lin}^t}{z_{o(kh)_lin}^t} \right) \right]}{\sum_{k=1}^{K} W^k \left[ 1 + \frac{1}{r_{1k} + r_{2k} + ngood_k} \left( \sum_{r=1}^{r_{1k}} \frac{s_{rokgood}^{t+}}{y_{rokgood}^t} + \sum_{r=1}^{r_{2k}} \frac{s_{rokbad}^{t-}}{y_{rokbad}^t} + \sum_{k_l}^{ngood_k} \frac{s_{ok_lgood}^{(t,t+1)}}{z_{ok_lgood}^{(t,t+1)}} \right) \right]} \tag{8}$$

ii. Division efficiency.

$$\varphi_0^* = \min \frac{\sum_{t=1}^{T} W^t \left[ 1 - \frac{1}{m_k + linkin_k + ninput_k} \left( \sum_{i=1}^{m_k} \frac{S_{iok}^{t-}}{x_{iok}^t} + \sum_{(kh)_l=1}^{linkin_k} \frac{s_{o(kh)_lin}^t}{z_{o(kh)_lin}^t} + \sum_{k_l}^{input_k} \frac{s_{ok_linput}^{(t,t+1)}}{z_{ok_linput}^{(t,t+1)}} \right) \right]}{\sum_{t=1}^{T} W^t \left[ 1 + \frac{1}{r_{1k} + r_{2k}} \left( \sum_{r=1}^{r_{1k}} \frac{s_{rokgood}^{t+}}{y_{rokgood}^t} + \sum_{r=1}^{r_{2k}} \frac{s_{rokbad}^{t-}}{y_{rokbad}^t} \right) \right]} \tag{9}$$

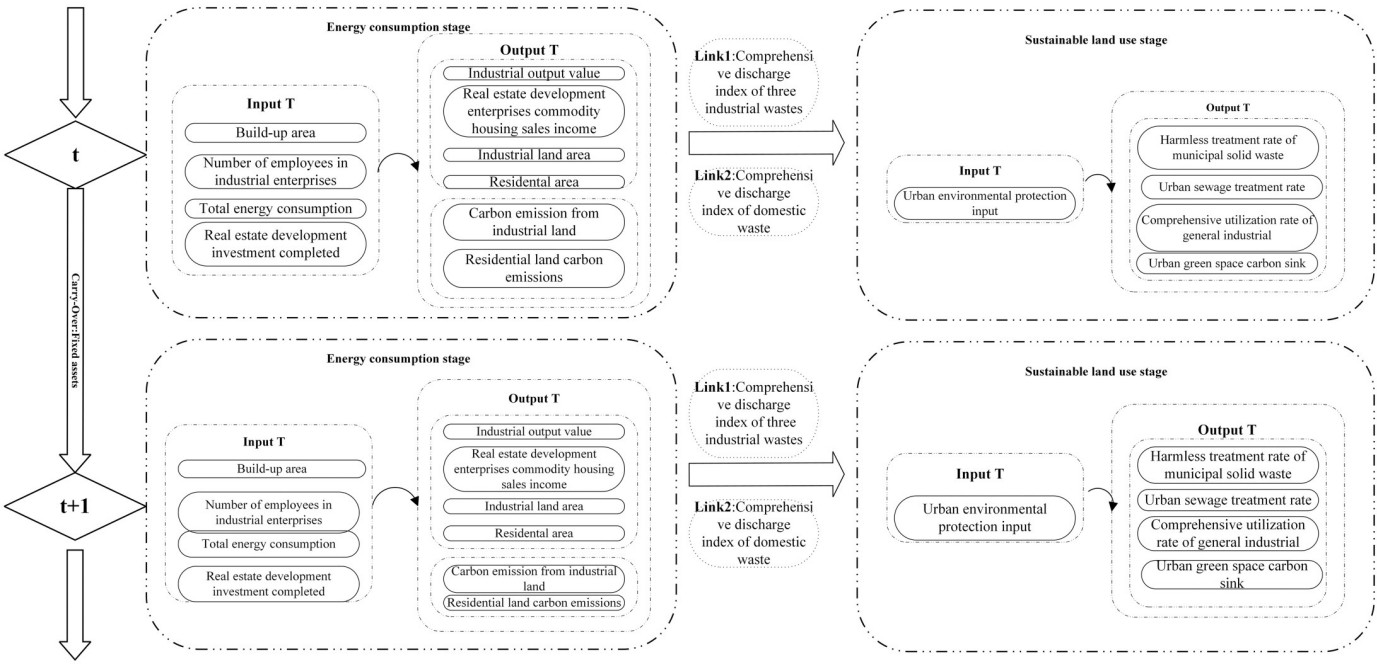

**Fig 1. TCEI assessment framework.**

Division period efficiency.

$$\rho_0^* = \min \frac{1 - \frac{1}{m_k + linkin_k + ninput_k} \left( \sum_{i=1}^{m_k} \frac{s_{iok}^{t-}}{x_{iok}^t} + \sum_{(kh)_l=1}^{linkin_k} \frac{s_{o(kh)_l in}^t}{z_{o(kh)_l in}^t} \sum_{k_l}^{input_k} \frac{s_{ok_l input\ input}^{(t,t+1)}}{z_{ok_l input}^{(t,t+1)}} \right)}{1 + \frac{1}{r_{1k} + r_{2k}} \left( \sum_{r=1}^{r_{1k}} \frac{s_{rokgood}^{t+}}{y_{rokgood}^t} + \sum_{r=1}^{r_{2k}} \frac{s_{rokbad}^{t-}}{y_{rokbad}^t} + \right)}$$  (10)

(c) Input, desirable output, and undesirable output efficiencies

To address any potential biases in traditional efficiency indicators, this study uses the total-factor energy efficiency index. This index includes eight crucial efficiency models: built-up area, total energy consumption, industrial land area, residential land area, carbon emissions from industrial land, carbon emissions from residential land use, urban environmental protection inputs, and urban green space carbon sink. Efficiencies are equal to 1 when the planned inputs align with the actual inputs, indicating complete efficiency. However, when the planned inputs are less than the actual inputs, efficiencies fall below 1, denoting overall inefficiency.

When the target desirable outputs match the actual desirable outputs, efficiencies are equal to 1, signifying complete efficiency. Conversely, when the target desirable outputs exceed the actual desirable outputs, efficiencies fall below 1, indicating overall inefficiency. The structure and variables of the intertemporal efficiency scheme, as well as the dynamic model's variables within the measurement framework, are depicted in Fig 1.

## 3.2 Tobit regression model

This study utilizes green space area, population respiratory carbon emissions, domestic waste removal, and general industrial solid waste emissions as explanatory variables. The efficiency

of carbon emissions from land use is as a dependent variable within the equation:

$$y_i^- = \beta_o + \sum_{k=1}^{K} \beta_k x_{ki} + \varepsilon_i \tag{11}$$

$$\left[ i = 1, 2, \cdots\cdots, n; k = 1, 2, \cdots\cdots, K; \varepsilon_i \sim N(0, \sigma^2) \right]$$

$$\begin{cases} y_i = y_t^\Gamma (y_i^\Gamma > 0) \\ y_i = 0 (y_i^\Gamma \leq 0) \end{cases}$$

The dependent variable $y_i$ indicates the efficiency of carbon emissions from land use, $x_{ki}$ is the explanatory variable, $\beta_0$ is the constant term, $\beta_k$ is the vector of regression parameters, and $\varepsilon_i$ is the residual term. The equation is characterized by the fact that the explanatory variable $x_{ki}$ is the actual observed value, while the explained variable $y_i$ can only be observed in restricted form. When $y_i^\Gamma > 0$, $= y_i$, $y_i^\Gamma$ and $y_i$ are unrestricted observations; when $y_i^\Gamma \leq 0$, $y_i$ is said to be a restricted observation.

### 3.3 LSTM prediction model

The LSTM model is a unique kind of RNN. LSTM is composed of several units that are in charge of holding data. Every storage component manipulates and processes data through four control mechanisms: forgetting gate, input gate, memory unit, and output gate. The important framework of LSTM appears in Fig 2. Here, $x_t$ denotes the input of parameter information used for model training, such as urban green space area, population respiration carbon emission, etc. $h_t$ and $h_{t-1}$ denote the outputs of LSTM in current and previous iterations, respectively. The forget gate determines the amount of information from the previous state that is kept or forgotten in the current state in $h_{t-1}$, where $C_t$ and $C_{t-1}$ represent the states of the storage unit and the results of model learning in the present and the last cycles, respectively. It means that LSTM uses these states to learn from the sequence of data over time with the ability to keep or forget information as needed. In the end, the output gate manages the release of valid data and rectifies the mistakes over a number of iterations. Oblivion gate $f_t$ determines how much information from the previous cell is retained by the current cell with the

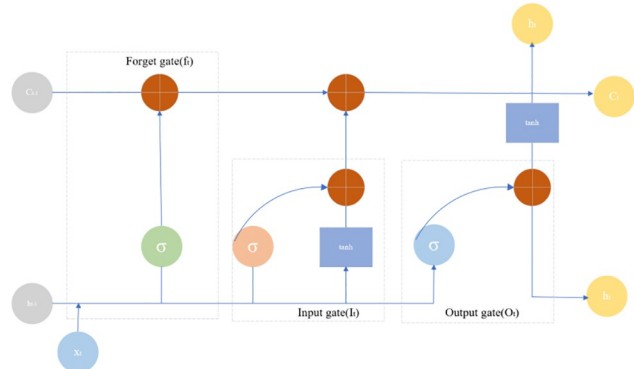

**Fig 2. Flowchart of LSTM neural network model operation.**

expression.

$$f_t = \sigma\left(w_f \cdot [h_{t-1}, \beta_m] + b_f\right) \tag{12}$$

Here, $w_f$ is the weight matrix, $b_f$ is the bias, and σ is an s-type function.

The input gate is responsible for determining the percentage of current input data that will be saved in the memory cell. This value is illustrated in Eq (13), in which the weight matrix $w_i$ and bias $b_i$ are represented. Concurrently, Eq (14) enables the derivation of new state information, with the weight matrix $w_c$ and bias $b_i$ again playing a role. Consequently, the current state $C_t$ of the learning outcome can be deduced from Eq (15).

$$i_t = \sigma(w_i \cdot [h_{t-1}, x_t] + b_i) \tag{13}$$

$$D_t = tanh(w_c \cdot [h_{t-1}, x_t] + b_c) \tag{14}$$

$$C_t = f_t \cdot C_{t-1} + i_t \cdot D_t \tag{15}$$

The output gate $O_t$ controls the state output from the memory cell, as shown in Eqs (16)–(17), where the weight matrix $w_0$ and bias $b_0$ are present. LSTM, with its four control gates and memory cells, is designed to read, refresh, and modify long-term information.

$$O_t = \sigma(w_0 \cdot [h_{t-1}, x_t] + b_0) \tag{16}$$

$$h_t = O_t \cdot tanh(c_t) \tag{17}$$

## 4 Empirical analyses

### 4.1 Data overview

This research selects 27 cities in the YRD Urban Agglomeration (2011–2020) as the study sample (Fig 3). YRD is not just a region with a robust foundation for urbanization in China, but also a significant international gateway in the Asia-Pacific region. This makes it an ideal case study for examining the carbon emissions resulting from land use.

Because using all the data from a specific region can make it challenging to accurately represent the true state of variables, this study employs per capita data for analysis to enhance its scientific relevance and precision. The data are sourced from the China Statistical Yearbook (2011–2020), China Urban Statistical Yearbook (2011–2020), and China Energy Statistical Yearbook (2011–2020). DEA was first proposed by Charnes (1979), a famous American researcher in operations research, to assess the relative effectiveness of DMUs. S1 Appendix lists the inputs, links, and outputs of the dynamic two-stage DEA model. This study sets built-up area, number of employees in industrial enterprises, number of real estate employees, total energy consumption, and real estate development investment completed as inputs for S1 (energy consumption stage) variables. It uses industrial output value, real estate development enterprises, commodity housing sales income, industrial land area, residential land area, carbon emissions from industrial land, carbon emissions from residential area, and other variables as outputs for S1. It then chooses comprehensive discharge index of three industrial wastes and comprehensive discharge index of domestic waste as link variables for both stages and at the same time identifies urban environmental protection inputs as the input variable for S2 (land sustainable use stage). Comprehensive utilization rate of general industrial solid waste, harmless treatment of municipal domestic waste, urban sewage treatment rate, and

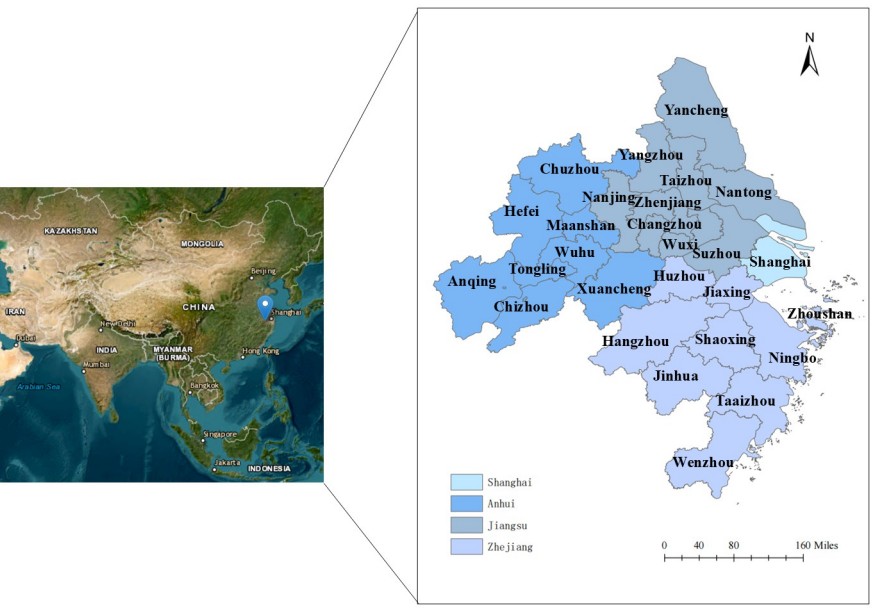

**Fig 3. Overview of the study area.** Notes: The images are from an open data platform. USGS EROS (Earth Resources Observatory and Science (EROS) Center) (public domain): http://eros.usgs.gov/#.

urban green space carbon sink are output variables of S2. Fig 4 presents the descriptive statistics for each variable.

## 4.2 Descriptive statistics

At two different stages, this study chooses 8 representative variables, computes their maximum, minimum, mean, and standard deviation, and rounds off the outcomes to two decimal places without altering the original meaning. See Figs 4 and 5 for details. (Fig 4a) illustrates with the acceleration of urbanization that the statistics of built-up area show a steady increase year by year, which is consistent with total energy consumption (Fig 4b). Industrial land area (Fig 4c) and residential land area (Fig 4d) both exhibit a notable decline following 2015—a trend that is strongly associated with the then-implemented policy aimed at curbing the excessive development of real estate and industrial land. On the other hand, the average quantity of carbon emissions from industrial land (Fig 4e) and residential land carbon emissions (Fig 4f) have been changing more smoothly. This suggests that the carbon emission level of the associated industries remains relatively consistent year after year.

Unlike Stage 1, the urban environmental protection inputs (Fig 5a) in Phase 2 are more volatile. On the one hand, their extreme value in 2011 is much higher than in other years. Conversely, substantial disparity exists between the extreme value and the comparably small and median values. This suggests that investment in environmental protection varies greatly across years and regions. In addition, the extreme value of urban green space carbon sink (Fig 5b) decreases in fluctuation, but the mean and standard deviation are basically the same.

## 4.3 Empirical result analysis

**4.3.1 Overall efficiency analysis.** Fig 6 shows the overall efficiency scores of the above 27 cities for the two stages (2011–2020). The total efficiency values of the 27 cities average 0.6 during the study period, and TCEI is significantly unbalanced among different regions.

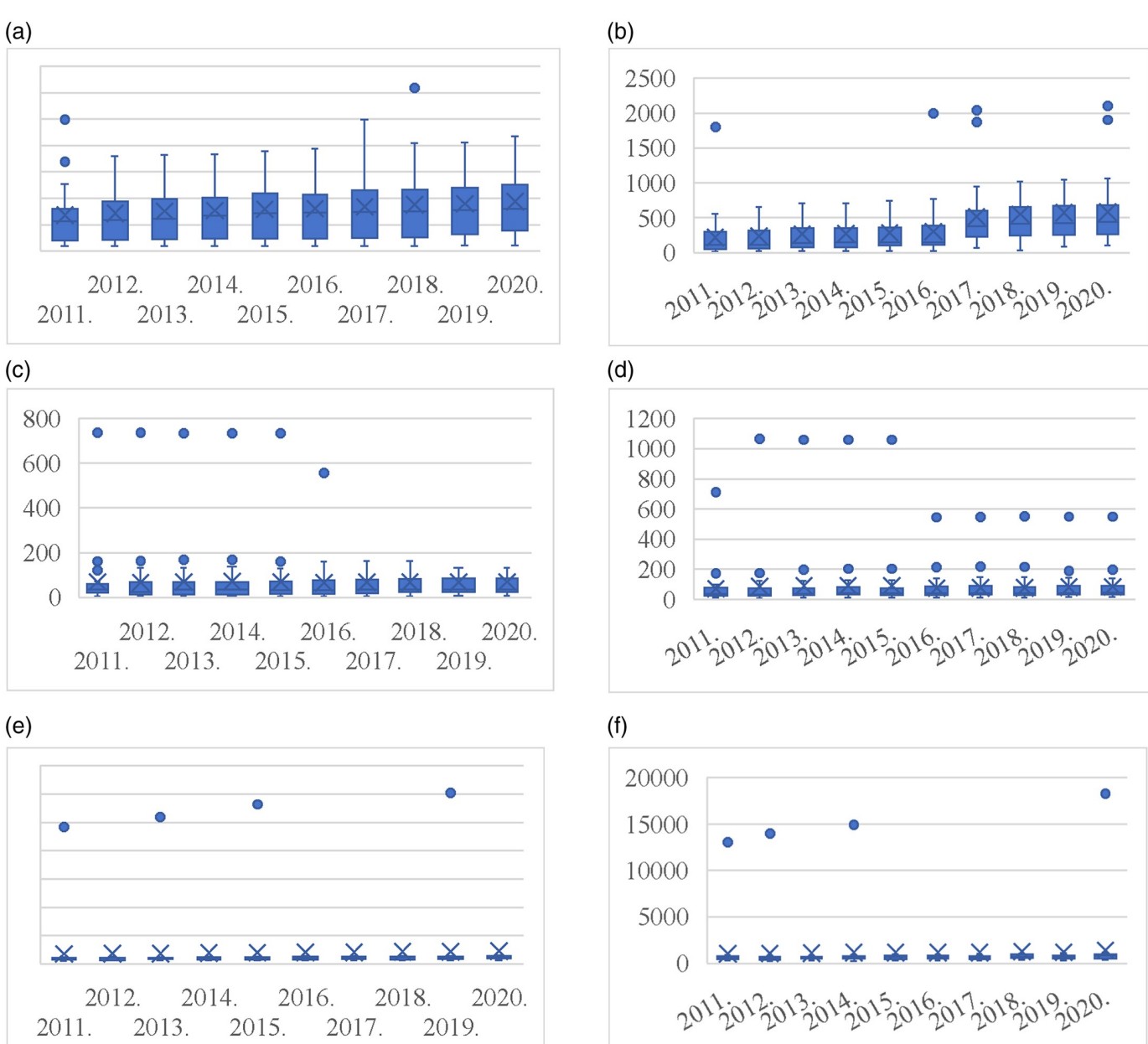

**Fig 4. Statistical analysis of input and output variables of Stage 1.** (a) Built-up area; (b) Total energy consumption; (c) Industrial land area; (d) Residential land area; (e) Carbon emissions from industrial land; (f) Carbon emissions from residential land use.

Specifically, Anhui has the largest TCEI, followed by Jiangsu. Zhejiang and Shanghai have the lowest TCEI. In addition, TCEI of the YRD urban agglomeration generally indicates an increasing trend within the study interval during this time period.

The study's findings reveal two key pieces of information. First, the cities of Chizhou and Wuhu in Anhui have higher TCEI values at 0.9516 and 0.9750, respectively. The bottom ranked cities are Huzhou and Jinhua in Zhejiang at 0.4025 and 0.4081. Chizhou and Wuhu, as two cities with low population density in Anhui, indicate a negative correlation between population agglomeration and carbon emission efficiency. Due to their mountainous location and

(a)

(b)

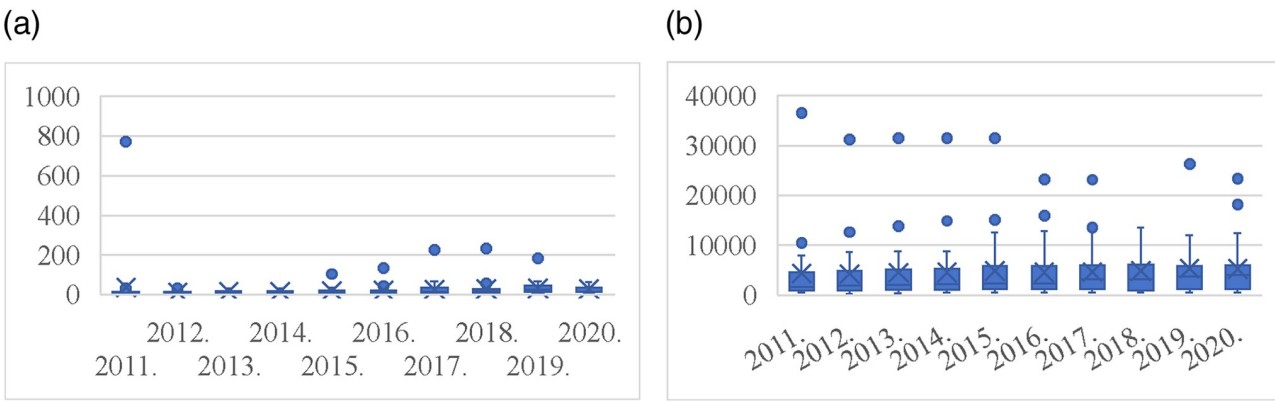

**Fig 5. Statistical analysis of input and output variables of stage 2.** (a) Urban environmental protection inputs; (b) Urban green space carbon sink.

narrow urban built-up areas, Huzhou and Jinhua have neglected the construction of urban green spaces, resulting in their own low carbon emission efficiency. Second, TCEI of most cities ranges from 0.5 to 0.8, with cities such as Nanjing and Hefei having relatively high TCEIs. This suggests that they are more efficient in land conservation and intensive use.

Fig 6 presents that TCEI of the YRD city cluster generally shows an increasing trend within the study interval. Specifically, the cities in Anhui and northern Jiangsu show a larger increase, especially Anqing, which rises from 0.5308 at the beginning to 0.8872 later. It may be because these latecomer cities, in the stage of new urbanization, timely changed their original extensive land use patterns, abandoned the initial concept of promoting economic development with heavy industry as the pillar industry, and carried out corresponding industrial transformation, thus accelerating the construction of green spaces in new urban areas and thereby reducing carbon emissions under the same energy consumption.

Some cities' TCEI in the intermediate urbanization stage has improved in a fluctuating state. The trend of change in their TCEI was relatively flat before 2015, while the efficiency has gradually increased in the years thereafter. For example, the efficiency values of cities in Zhejiang vary between 0.3–0.5, which suggests that there has not been any significant enhancement in the overall land-use efficiency of these cities, implying ample room for improvement. As for the more economically developed areas such as Shanghai, Hangzhou, and Nanjing, their TCEI shows a clear downward trend. For example, Shanghai declined from an efficiency value close to 1 at the beginning to 0.4565 later. This may be due to the rapid urban expansion stage, where the government converted more and more urban green space into construction land to promote economic development, thus greatly reducing the carbon sequestration capacity of green space.

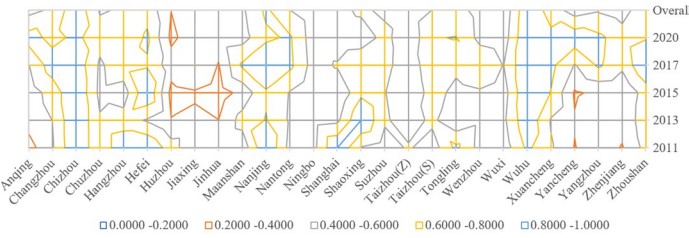

**Fig 6. Changes in overall efficiency by city for representative years.**

**Table 1. Changes in the efficiency of energy consumption stages by provinces in YRD, 2011–2020.**

| Province | Mean | 2011 | 2012 | 2013 | 2014 | 2015 | 2016 | 2017 | 2018 | 2019 | 2020 |
|---|---|---|---|---|---|---|---|---|---|---|---|
| Shanghai | 0.7218 | 1.0000 | 0.6292 | 0.3297 | 0.5613 | 0.3835 | 1.0000 | 1.0000 | 0.9718 | 0.6642 | 0.6779 |
| Anhui | 0.7614 | 0.6744 | 0.7073 | 0.6683 | 0.6786 | 0.7014 | 0.8230 | 0.8740 | 0.9003 | 0.7754 | 0.8108 |
| Jiangsu | 0.8352 | 0.7072 | 0.6957 | 0.8201 | 0.7340 | 0.7533 | 0.8827 | 0.9593 | 0.9743 | 0.9093 | 0.9166 |
| Zhejiang | 0.5695 | 0.5176 | 0.4611 | 0.5599 | 0.4782 | 0.4480 | 0.6403 | 0.7187 | 0.7324 | 0.5719 | 0.5668 |

**4.3.2 Stage efficiency analysis.** *4.3.2.1 Energy consumption stage (S1).* This study now delves into the complexity of efficiency in these two stages. As Table 1 shows, Jiangsu demonstrated commendable efficiency in the energy consumption stage from 2011 to 2020 with an average value of 0.8352, which is impressive. This number rises, possibly due to excessive restrictions imposed by the province on converting arable land into construction sites during the urbanization process. The province has also been actively replacing traditional energy with cleaner alternatives. The slow population growth caused by limited urban size has further promoted the improvement of energy consumption efficiency in Jiangsu's cities.

Zhejiang has the least efficient carbon emissions in terms of overall land usage, averaging 0.5695. The lack of significant fluctuation during the study period suggests that, when considering the carbon emission factor, the overall efficiency of land resource allocation in Zhejiang is low. On the other hand, TCEI of land usage in Shanghai and Anhui is higher, averaging 0.7218 and 0.7614, respectively.

The changes shown in Table 1 illustrate from 2011 to 2013 that the carbon emission efficiency of land use in Shanghai significantly decreased, and the fluctuations were relatively small thereafter. This may be because Shanghai's urban expansion became increasingly rapid in the two years after 2011, and by 2013 its urbanization process had basically ended. Out of the four regions, Jiangsu has the highest land-use carbon emission efficiency value, peaking at 0.9743 in 2018 and hitting a low of 0.6957 in 2012. The reason is that Jiangsu's strict ecological public welfare forests occupy and compensate for the balance of measures to ensure the carbon sink function of green space. At the same time, the government carried out strict total construction land use and intensity control after 2018, which slowed down the process of the original natural land surface, such as arable land, flowing into construction land. The highest efficiency value of land use TCEI in Anhui also appeared in 2018 at 0.9003, and the efficiency value fell to 0.7014 in 2015 and thereafter to the lowest value of 0.6744 in 2011. This may be due to the optimization and adjustment of the industrial structure of Anhui after 2018, replacing the original manufacturing industry and heavy chemical industry with high value-added secondary and tertiary industries. Finally, among the four regions, Zhejiang recorded the least overall efficiency value and exhibited the most minimal change. Notably, its peak efficiency was in 2018, achieving a value of 0.7324, while its lowest efficiency occurred in 2012 at a value of 0.4611. This could be attributed to the recent slump in the real estate market, which resulted in unutilized construction land.

*4.3.2.2 Land sustainable use stage (S2).* In the second stage (Fig 7), Anhui performed the best in terms of efficiency in the sustainable land use stage from 2011 to 2020, maintaining a stable average of 0.7127 and continuously improving every year. The reason is that due to a certain gap in the total GDP of Anhui compared to Jiangsu, Zhejiang, and Shanghai, Anhui's carbon emissions level is lower than that of Jiangsu, Zhejiang, and Shanghai. The financial investment related to environmental protection is similar among different cities, and the reduction of energy consumption waste and garbage caused by population outflow has led Anhui to invest more in urban green space construction.

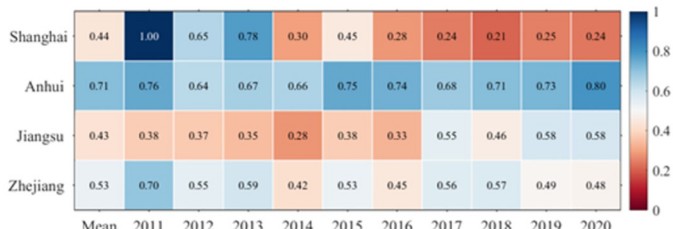

**Fig 7. Changes in the efficiency of land sustainable use stages by province in the Yangtze River Delta region, 2011–2020.**

The average value of land sustainable use stage in Jiangsu is 0.4260. Throughout the entire study period, the efficiency of sustainable land use in Jiangsu was the lowest, fluctuating between 0.3 and 0.6. This can be attributed to the rapid development of urbanization and the subsequent increase in energy consumption there. At the same time, the influx of a large number of people has forced environmental protection financial investment to tilt towards household waste management, thereby occupying the financial space for green space construction. The sustainable land use efficiency of Shanghai and Zhejiang is ranked according to their population density with average values of 0.4392 and 0.5335, respectively.

From the changes in efficiency values during the research period (Fig 7), Anhui's land use carbon efficiency value is the highest among the four regions and has almost maintained an increasing trend year by year. Among the four regions, Zhejiang ranks second in overall efficiency value and has the most stable changes. This indicates that Anhui and Zhejiang have relatively coordinated management of urban carbon emissions. These local governments have effectively balanced the development goals of urbanization and ecological protection, timely increased investment in urban ecological restoration, and thus enhanced the carbon sequestration role of urban land.

During the research period, the efficiency of sustainable land use in Shanghai has been decreasing year by year, from reaching a peak efficiency value of 1 in 2011 to a minimum efficiency value of 0.24 in 2020. Among these four regions, Jiangsu has the lowest average efficiency value during this stage. The lowest value of sustainable land use efficiency in the province was 0.2801 in 2014, while its peak value was 0.5847 in 2019. This trend closely relates to the regional agglomeration characteristics of population and industry in Jiangsu. Especially in some cities in northern Jiangsu that are still undergoing urbanization, the urban industrial structure continues to be mainly dominated by high polluting heavy industry, and the transition to more environmentally friendly industries such as the tertiary industry will be a long process.

**4.3.3 Analysis of important input-output indicators.** As depicted in Fig 8, there is a significant variation in the efficiency performance of the primary input-output indicators from 2011 to 2020. Due to space limitations, the efficiency values of all input-output indicators are not listed. However, this research does analyze the key indicators mentioned above in two stages.

Regarding the trend of changes in the indicator of built-up area (Fig 8a), Shanghai has the highest built-up area efficiency value of the three regions. It peaked at 1 in 2019, while the least efficient year was 2013 when it reached 0.4959. Shanghai is the city with the largest built-up area in eastern China. Its urbanization rate at the end of 2021 was as high as 89.3%, exceeding the average level of developed countries and entering the late stage of urbanization development. Therefore, the contribution of the built-up area indicator is larger in Shanghai. Anhui

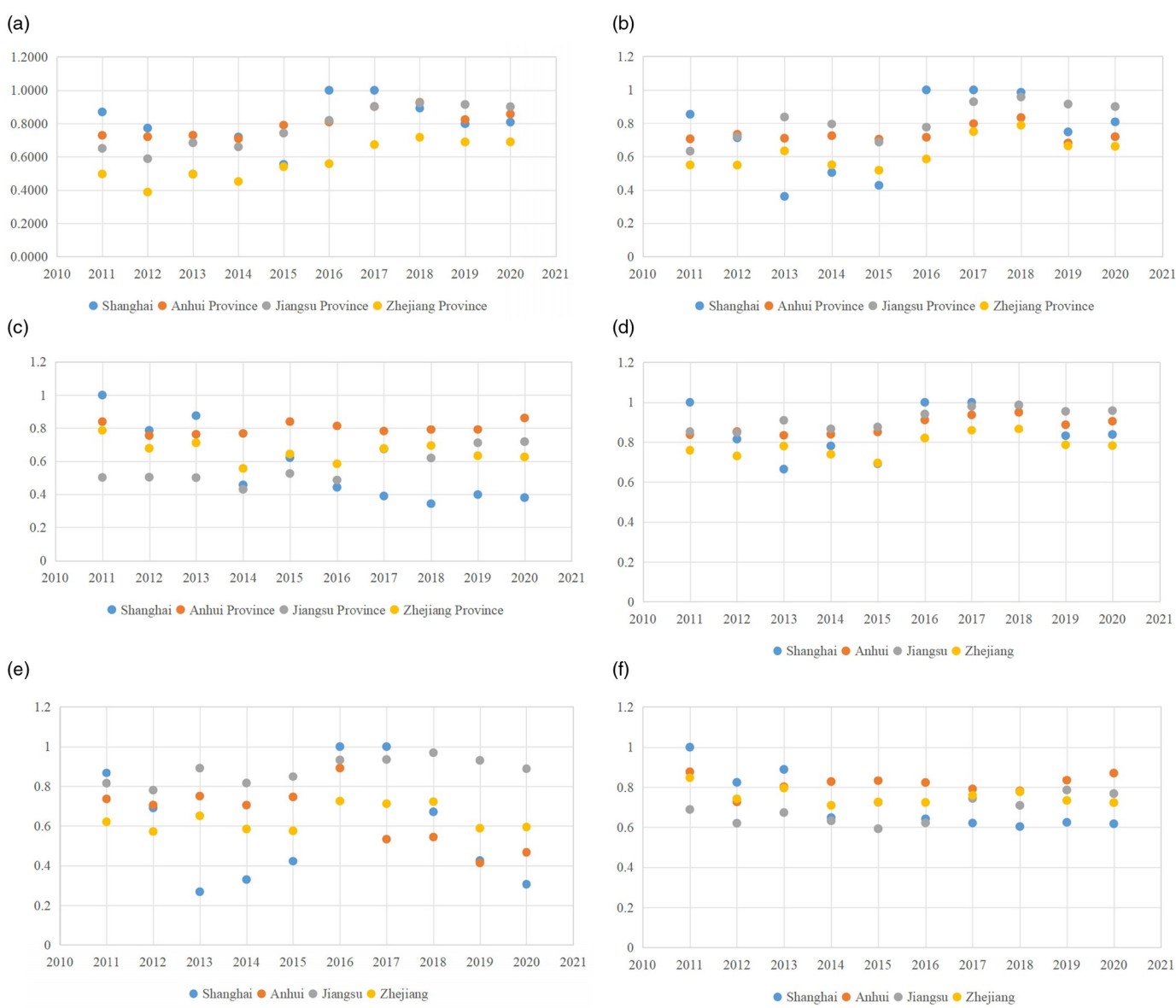

**Fig 8. Efficiency of important input-output indicators.** (a) Built-up area; (b) Total energy consumption; (c) Industrial output value; (d) Carbon source of construction land; (e) Urban environmental protection inputs; (f) Urban green space carbon sink.

ranks second in performance. Although its built-up area efficiency value dropped after hitting 0.9277 in 2018, the province's overall performance remains stable with potential for enhancement. Conversely, Zhejiang recorded the lowest efficiency with yearly fluctuations. The peak year was 2018 with an efficiency value of 0.7107, while the least efficient year was 2012 with a value of 0.3879.

Regarding the trend of changes in the indicator of total energy consumption (Fig 8b), the overall efficiency of the four regions is above 0.6 every year, with Jiangsu having the highest average value during the study period at 0.8149. Shanghai ranks second, reaching a peak of 1 for two consecutive years after 2017. There are five years in which the efficiency value exceeded 0.8. Anhui ranked third, with peak efficiency of total energy consumption occurring in 2018 at

0.8344. The lowest value appeared in 2019 at 0.6805 with little fluctuation over the study period. Zhejiang performs slightly worse, with two years above 0.7 and a minimum value of 0.5180, and fluctuates between high and low values. Its development goal of promoting a short-term economy will inevitably increase the consumption of traditional energy sources. After 2017, the region represented by Jiangsu increased the development and construction of presumably cleaner energy sources, such as hydropower, thereby accelerating the process of cleaning up the energy supply, which in turn leads to lower carbon emissions. In order to achieve the goal of carbon neutrality, the energy production structure of other cities in the YRD cluster has gradually shifted from coal-based to diversified renewable energy sources.

Regarding the trend of changes in the indicator of industrial output value (Fig 8c), the industrial output value of Shanghai experienced a significant decrease from 1 in 2011 to 0.3807 in 2020. This steep decline implies that its industrial output value is highly unstable, and there is considerable scope for improvement in the future. The efficiency of industrial output value in Anhui is consistently greater than 0.7 and fluctuates insignificantly during the study period. The highest efficiency value is 0.8606 in 2020, and the lowest is 0.7550 in 2012. The highest efficiency year in Jiangsu is 0.7179 in 2020, and the lowest is 0.4303 in 2014. The performance for the remaining years remains stable, fluctuating between 0.45 and 0.68. The value of industrial output value in Zhejiang shows a decreasing trend in fluctuation, with its highest value decreasing from 0.7863 at the beginning to 0.6259 later on. Overall, there are relatively few regions where industrial output value efficiency reaches a desirable value or shows an upward trend. This indicates in most regions that carbon emission efficiency of industrial output is not satisfactory, and there is much room for optimization.

Regarding the trend of changes in the indicator of carbon source of construction land (Fig 8d), the best-performing province for this indicator is Jiangsu, with the highest value occurring in 2018 at 0.9872 and with six years of efficiency values above 0.9. The second best-performing region is Anhui, with its maximum value appearing in 2020 at 0.8606, and the other years fluctuating around 0.8 with a more stable trend of change. The next best performer is Shanghai, with the indicator reaching its maximum value in 2011, 2016, and 2017. A worse performer is Zhejiang, with its maximum value appearing in 2018 at 0.8662 and fluctuating within the range of 0.69–0.85 in the other years. The swift expansion of construction land in the YRD urban agglomeration has resulted in a quick surge in carbon sources. Consequently, the intensity of carbon emissions from land use has escalated rapidly. Hence, as its economy grows rapidly, the YRD urban agglomeration must shoulder a greater share of emission reduction responsibilities and stringently regulate the unchecked expansion of construction land.

Regarding the trend of changes in the indicator of urban environmental protection inputs (Fig 8e), Jiangsu performs the best and reaches a maximum efficiency value of 0.9695 in 2018, but also shows a decreasing trend from 0.9309 in 2019 to 0.8885 in 2020. Zhejiang is more stable, with efficiency values ranging from 0.5722 to 0.7252 and reaching the optimal efficiency in 2016. Shanghai has a large difference between different years. Its efficiency value reaches 1 in 2016 and 2017 and later drops to 0.3059 in 2020. This reflects the regional heterogeneity of urban environmental protection investment intensity. It is because the rise in investments towards urban environmental protection projects can enhance the conservation and intensification of land within the YRD urban agglomeration. It can also counteract the unchecked expansion of construction land and thereby boost overall efficiency of carbon emissions.

Regarding the trend of changes in the indicator of urban green space carbon sink, the efficiency values of all regions perform well with an average value of 0.6837 or above (Fig 8f). Among them, the best performance is in Anhui with the highest quantity of 0.8777 in 2011 and the lowest value of 0.7269 in the next year, after which it has been fluctuating steadily at the level of 0.8 or below. This suggests that the regions have increased their investment in

urban forest construction in recent years, and so the increase in green space has improved the overall carbon sequestration in the region.

**4.3.4 Analysis of factors affecting carbon emission efficiency.** The efficiency of carbon emissions in the land use process is intimately linked to a city's industrial structure, urban population density, and level of urban environmental governance. Due to the dynamic two-stage DEA model generating efficiency values between (0, 1), which are constrained dependent variables, the Tobit model is chosen to conduct a truncated regression analysis. This analysis deeply explores the factors influencing TCEI and the degree of their impact. Four indicators related to national production and daily life have been selected, based on relevant literature [37–39].

The names and definitions of the influencing factors appear in Table 2. In the Tobit model, these four influencing factors are referred to as technical non-efficiency factors. The carbon emission efficiency influencing factor model is constructed by combining these with the Tobit model.

According to the calculation results in the previous section, Eq (5) is solved via Stata2019 software, and the results are in Table 2. First, the results imply that each variable has a significant effect on carbon emissions in the process of land use. Four variables—namely, green space area, population respiratory carbon emission, domestic rubbish removal, and general industrial solid waste emission—are significant at the 1% level. Second, the regression coefficients of the two variables of carbon emissions from human respiration and domestic rubbish removal are significantly negative with t-values of -4.11 and -5.53, respectively, indicating that these two drivers negatively correlate with TCEI of land use, and that the agglomeration effect of the population brought about by the urbanization of the land increases the pressure of urban carbon emissions. This also corresponds to the results of efficiency analysis. Finally, urban green space area and general industrial solid waste emissions have a significantly positive impact on urban land TCEI enhancement. The t-values of these two variables stand at 4.80 and 6.19, respectively, suggesting that an increase in the area of green space significantly enhances the carbon sequestration capacity of urban green spaces. This in turn aids in achieving the goal of reducing carbon emissions. The environmentally friendly treatment of industrial solid waste is easier and reduces the degree of industrial carbon emissions. At the same time, the growth of urban industrial solid waste also represents the growth of urban economic level, thus comprehensively improving the carbon emission efficiency of urban land use from two aspects.

**4.3.5 Forecasts of TCEI for the YRD urban agglomeration over the next six years.** The LSTM neural network model is employed to forecast the four drivers of carbon emission efficiency over a rolling window, ultimately deducing and estimating the carbon emission efficiency values of the YRD city cluster for the period 2021–2026. To avoid some problems like

**Table 2. Parameter estimation results of the impact factor model.**

| Variable | Symbol | Unit (of measure) | Coefficient | t | P>\|t\| |
|---|---|---|---|---|---|
| Green area | Z1 | Square kilometer | 0.702*** | 4.80 | 0.000 |
| Carbon emissions from human respiration | Z2 | tons | -0.806*** | -4.11 | 0.000 |
| Domestic garbage removal volume | Z3 | tons | -1.302*** | -5.53 | 0.000 |
| General industrial solid waste emissions | Z4 | tons | 0.300*** | 6.19 | 0.000 |

Note:

*** represents a significance level of 1%, while

** represents a significance level of 5%.

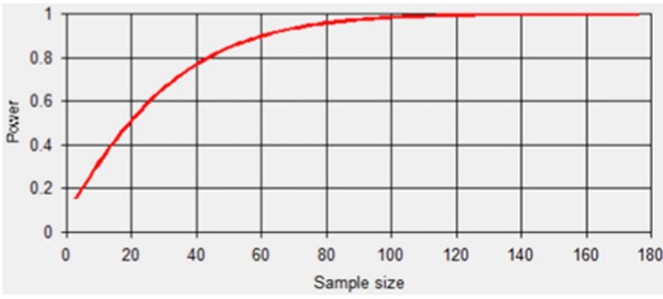

**Fig 9. Graph of statistical power and sample size test results.**

the size of the effect being not practical, an effect size test is conducted to determine the rationality of the sample size. Moreover, the statistical efficacy result is 0.804, which suggests that our experimental and control data are reasonable. Fig 9 presents a statistical power graph based on the calculation results.

The time series data, which include the four selected variables and TCEI, are standardized to eliminate the difference in magnitude. The matrix, which is made up of the values from the initial t cycles of the land use carbon emission efficiency sequence, is used as input feature data for training in the LSTM neural network. The data value from the t+1$^{th}$ cycle is then the output for error-correction comparison. Since the total number of observed cities is 27, the sample capacity is small, and so we use a sliding time window to expand the sample set, forming seven sets of samples for 2011–2014, 2012–2015, to 2017–2020. The size of the implied layer of the constructed LSTM model is 1, the training step size is 1, the number of epochs is 400, the number of neurons is 270, and the optimizer selects Adam as shown in Table 3. This paper judges the prediction effect of the LSTM model based on $R^2$, MSE (Mean-square error), and MAE (Mean absolute error). The extended sample repeats the aforementioned process to obtain the predicted carbon emission efficiency values for the YRD city cluster for the period 2021–2026. The forecast implies that the carbon emission efficiency of the YRD cities is expected to fluctuate between 0.3 and 1.1 in the future.

After the LSTM model training is complete, the forward rolling window prediction method is used. This means during out-of-sample prediction the one-step prediction values obtained by using the in-sample data as feature data in the model are extended to the sample data for further application. The fitting results of the four input variables are in Table 3, which shows that the real and predicted values of the land use carbon emission efficiency of the training set of the cities fit well, as Fig 10 shows. This suggests that the LSTM model possesses robust generalization (Fig 10a), making it more apt for data with fewer observations like carbon emission intensity (Fig 10b). It can more effectively approximate any function with a certain level of generalization ability (Fig 10c), thereby ensuring reliability of the prediction (Fig 10d).

**Table 3. Evaluation index data of the fitting effect of each input-output index.**

| Variable | $R^2$ | adjusted_$R^2$ | MAE | MSE |
|---|---|---|---|---|
| Green area | 0.9763 | 0.9755 | 0.0115 | 0.0786 |
| Carbon emissions from human respiration | 0.9824 | 0.9818 | 0.0056 | 0.0548 |
| Domestic garbage removal volume | 0.9620 | 0.9607 | 0.0135 | 0.0765 |
| General industrial solid waste emissions | 0.9800 | 0.9792 | 0.0251 | 0.1239 |
| Carbon emission efficiency | 0.9752 | 0.9712 | 0.0139 | 0.0834 |

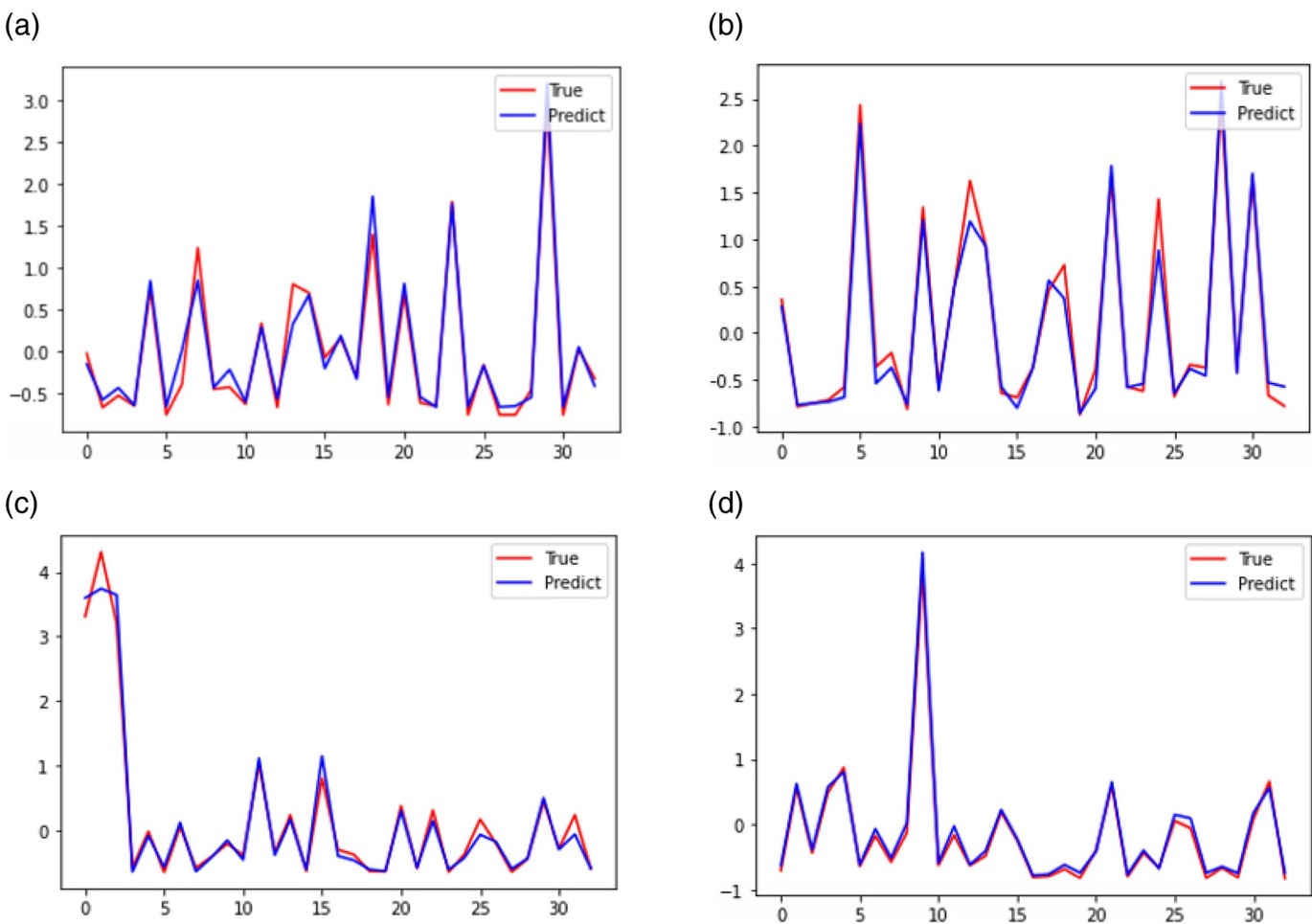

**Fig 10. Plot of input variable fit.** (a) Effect of fitting the green space area training set; (b) Effect of fitting the carbon emissions from human respiration training set; (c) Effect of fitting the training set on domestic waste removal; (d) Effect of fitting the training set on general industrial solid waste emissions.

In terms of the prediction of TCEI in the YRD agglomeration, the calculation is derived by using the in-sample data as feature data in the model and is extended to the sample data for further use. The test set of the better LSTM model for the land use carbon emission efficiency of the urban agglomeration of the YRD is $R^2 = 0.9752$, MAE = 0.0139, and MSE = 0.0834. The research further adopts the learning curve to determine whether there is overfitting phenomenon in the prediction of efficiency value. As shown in Fig 11, the training set and verification set both have lower loss errors. On the one hand, the training set loss gradually decreases and flattens with the increase of training samples, indicating that adding more training samples does not improve the performance of the model on training data. On the other hand, the learning curve of the training set has a high validation loss at the beginning, gradually decreases, and tends to be flat with the increase of training samples. The results imply that the majority of the indices fit well without any overfitting, demonstrating a certain degree of generalization capability that prediction reliability. Additionally, the model's high prediction accuracy confirms that the four input factors have a strong correlation with carbon emission efficiency.

The fitted curve of the projected carbon emission efficiency value for YRD is depicted in Fig 12. The findings indicate that the city with the highest land use carbon emission efficiency

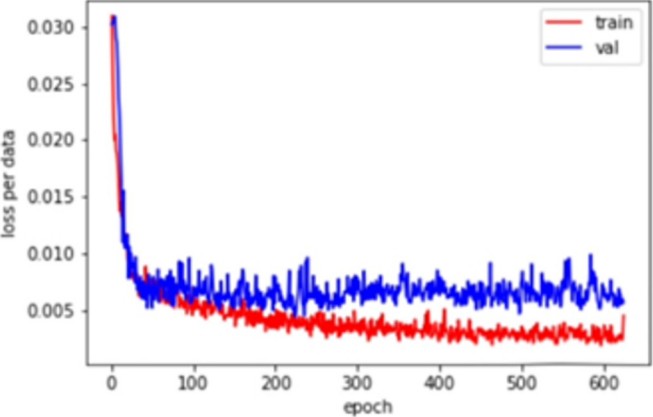

**Fig 11. Graph of training set and test set learning curve.**

in YRD could achieve an efficiency value of 0.9480 in the forthcoming six years. Overall, the forecasted efficiency values for the cities within the studied area are predominantly within the 0.65–0.75 range. Five cities have predicted values of 0.92 or higher, and these cities all have efficiency values of 1 in the training set. Moreover, the lowest land use carbon emission efficiency value is 0.3913, and the predicted efficiency value of the city is 0.06 higher than the real value of the training set.

## 4.4 Discussion

Taking YRD as the study area, a two-stage dynamic DEA, Tapio, and LSTM model are used to perform measurement and prediction of TCEI from the perspective of land use. The TCEI in

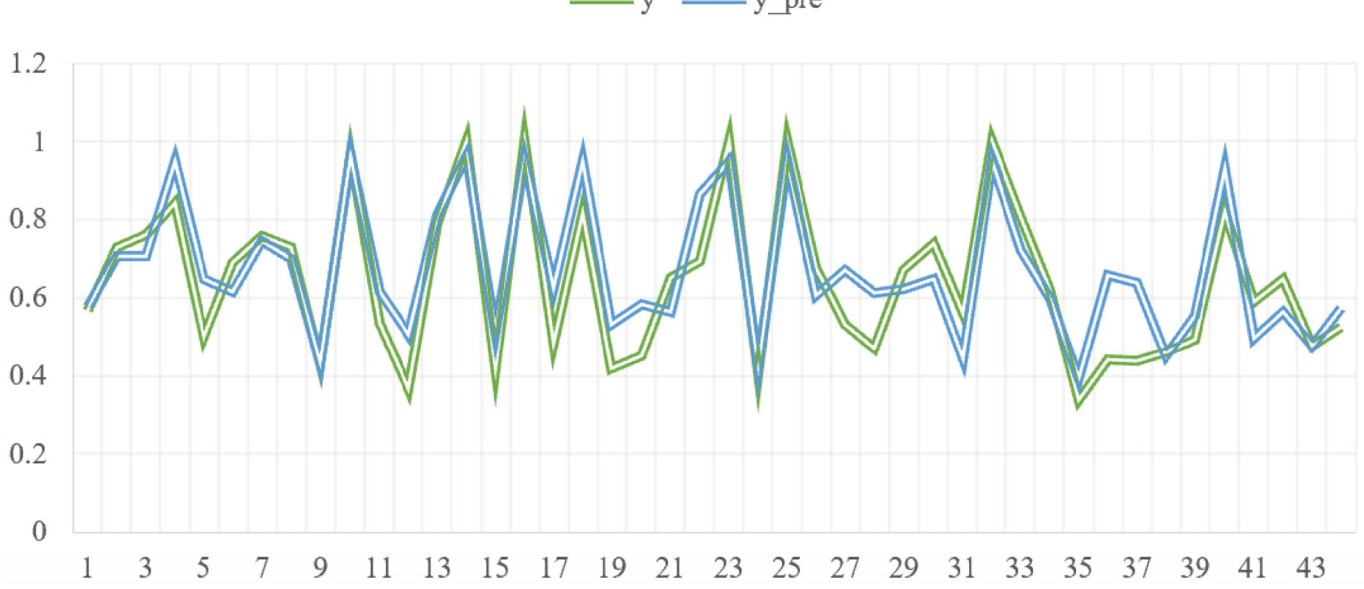

**Fig 12. Projected carbon emission efficiency of the Yangtze River Delta urban agglomeration in the next six years.**

each city of YRD during the study interval fluctuates, but it is generally at a high level. Nevertheless, it is still worthwhile to consider the heterogeneity of various stages and the underlying causes.

In terms of the TCEI of land use calculation, total efficiency has obvious spatial and temporal heterogeneity. It is not unexpected that provincial capitals, metropolitan areas, and coastal developed cities show higher TCEI. This is because as the provincial capitals of their respective provinces, although these cities have a relatively high population size and built-up area, they have already relocated heavy polluting industries and carried out a large amount of urban green space construction in conjunction with the existing suburban green spaces within their boundaries during the urban development process. Economically developed areas often have strong talent accumulation and resource endowment advantages, thus promoting the innovation of production technology and land use [40]. The cumulative effect of land resources and technological innovations in cleaner production allows these cities to reduce carbon emissions while maintaining productivity. Moreover, economically developed areas are better able to respond to and implement energy conservation and emission reduction policies [41]. In particular, the TCEIs of some cities in Anhui and northern Jiangsu have experienced significant increases, whereas the TCEIs of other cities have shown fluctuations. The carbon emission efficiency of land use in cities in southern Jiangsu is moderately high. The reason is that these economically developed areas often have a higher level of intensive land use. TCEI decreases in Zhejiang and Shanghai more substantially, suggesting significant potential for improvement in land use and resource allocation in these areas. Carbon emission performance varies across regions and different urbanization stages, demonstrating significant spatial and cyclical disparities.

In terms of stages, the efficiency values of most cities in the initial stage (energy consumption) exceeds the second stage (sustainable land utilization). In the initial stage, Zhejiang has the lowest total land use carbon emission efficiency among the four regions, but in the second stage, it has the second highest total efficiency value with smaller fluctuations. This result indicates that terrain factors may be the fundamental reason affecting energy consumption efficiency in cities. Zhejiang, which has the highest proportion of mountainous areas, needs to consume more energy in the process of urbanization, while Jiangsu, which is located in a plain area, does the opposite. These improvements are a result of China's long-term ecological preservation initiatives aimed at enhancing its cities' green development [42, 43]. In addition, among the six major input-output indicators, Jiangsu exhibits the best performance in terms of the indicators relating to carbon source of construction land and urban environmental protection input. This indicates with the advancement of urbanization and the development of industries in the province that the environmental problems in Jiangsu do not match the final level of economic development [44].

According to the forecast results of TCEI, there is a trend of further optimization and growth of TCEI in the next six years, which indicates that the Yangtze River Delta urban agglomeration in recent years has controlled the expansion of urban built-up areas and promoted the mixed-use development of central urban areas. This has significantly helped improve carbon emission efficiency. In other words, an intensive land use planning model has a positive impact on optimizing land use structure and the carbon emission reduction effect [45]. This highlights the increasingly significant squeezing phenomenon of urban green space caused by net population inflow after the expansion of urban built-up area in economically developed cities, such as Shanghai, and their land sustainable use efficiency has become the worst in the entire Yangtze River Delta region. The identification results of influence factors indicate that green space carbon sequestration capacity and population agglomeration effect are the core driving factors of urban land carbon emissions. Based on this, each region should

seek suitable land use methods according to its own characteristics so that its carbon emissions can reach an appropriate level.

From a research perspective, this study takes the comprehensive carbon emission efficiency of land use and focuses on assessing TCEI at different stages under the macro-level perspective. Land serves as an important carrier for urban economic development, while different types of land assume different functions in regional economic expansion [46, 47]. In addition, 27 cities in YRD are taken as case studies for this paper. There are many other studies focusing on carbon emissions throughout Asia as well as worldwide [48, 49]. Therefore, future research could include other developing countries' provinces or regions for analysis to show clearer characteristics and to help guide planning, management, and decision-making.

## 5 Conclusions and suggestions

### 5.1 Conclusions

This research employs a dynamic two-stage DEA model to build a TCEI assessment framework. Subsequently, the study determines TCEI in the land use process of YRD from 2011 to 2020 and further predicts their dynamic evolution. The findings suggest the following.

1. This study takes land use in the urban carbon emission efficiency index system as a new entry point. Looking at the total efficiency values, the average for the 27 cities during the study period is around 0.6, showing notable differences between groups. Specifically, Anhui has the highest carbon emission efficiency, followed by Jiangsu. Zhejiang and Shanghai have the lowest levels. During the study period, TCEI of the YRD city group generally shows an increasing trend within the study interval. This fills the first distribution identified in the 1 Introduction chapter.

2. In terms of phases, during the study period the efficiency value of the first phase for Shanghai maintained a high level of around 0.7. However, the second phase shows a significant fluctuation in land-use carbon emission efficiency, with a difference exceeding 0.75 between the highest and lowest efficiency values. It suggests that this type of highly developed economic region should take advantage of its own strengths to promote innovative research and develop energy technologies and emission reduction technologies. This phased DEA method overcomes the defects that the traditional DEA model cannot reflect, has higher accuracy and flexibility, and indicates the second distribution identified in the 1 Introduction section.

3. In terms of sub-indicators, the contribution of Shanghai to the built-up area indicator is larger. It shows that Shanghai mainly improves the level of urban development by increasing the area of built-up land. As a result, it is crucial to optimize the industrial structure and land resource allocation and to increase the environmental protection input in areas with inefficient land use and carbon emission. These findings make up for the shortcomings of other DEA models found in the 2 Literature review.

4. The results of driver identification of green space area and general industrial solid waste emissions on the efficiency of land use carbon emissions are all dominated by positive effect. Conversely, the population respiratory carbon emissions and the amount of domestic waste removal are dominated by negative effects. This illustrates the third distribution identified in the 1 Introduction section.

5. From the prediction results, this study selects the LSTM model and predicts that the land use TCEI of YRD will fluctuate in the range of 0.3–1.1 in the next six years. To some extent, it is better than the calculated results of the efficiency values of the two stages. Most of the

sample's predicted values fluctuate slightly higher than the efficiency value. This study will conduct future prediction research on the basis of carbon emission efficiency, reflecting another contribution to the fourth aspect.

## 5.2 Limitations

The panel data of YRD from 2011 to 2020 are employed to analyze LUEE and the driving factors in this study. The most recent year is currently unavailable due to the time gap in obtaining statistical data. Thus, future research may endeavor to establish a more comprehensive TCEI indicator system and increase the sample size by incorporating indicators that reflect the quality of life of residents and industrial upgrading factors in order to investigate additional factors that influence environmental efficiency in the context of urbanization.

Moving forward, it is important to investigate the effects of various land types on urban carbon emission efficiency using a macro-micro integrated approach. This should enhance our understanding of the role and influence mechanism of land use efficiency and the entire ecosystem [50]. The goal is to strengthen the symbiotic relationship between the healthy operation of ecosystems and the superior growth of the macro-economy.

## 5.3 Suggestions

Each region in YRD displays considerable variation in carbon emission efficiency, and there is a notable disparity in the land-use carbon emission performance of each city between the two phases. For this reason, cities with lower carbon emission efficiency should introduce advanced emission reduction and energy-saving technologies and management methods, focus on the green economy industrial belt with a high growth rate, and reduce regional efficiency differences. If YRD wants to keep the trend of decreasing carbon emissions and increasing TECI in the process of land use, then it must control the development and construction of construction land. In addition, the main way to reduce carbon emissions from carbon sources is to reduce industrial land and land area of high-energy enterprises, which requires YRD to concentrate on planning industrial parks and large buildings so as to further achieve its carbon emission reduction target [51, 52]. Therefore, the key to significantly improving the efficiency of sustainable land use in the region lies in concentrating the use of land according to the characteristics of industrial development to avoid urban land waste, while paying attention to the progress of low-carbon technology in industries and the construction of urban green spaces.

The contribution of regions to different indicators varies greatly depending on regional policies and development priorities. Local authorities should enforce pertinent laws and regulations, integrate green land use into government strategies, and introduce tax incentives for cleaner production along with carbon emissions trading schemes. Concurrently, the application of green technologies should be expedited, and waste recycling and reuse technologies should be employed to address issues of high energy consumption and pollution, thereby achieving sustainable development.

Local governments can narrow regional differences by formulating differentiated land-use policies and emission reduction pathways. At the same time, taking into account the spillover effect between different regions, policymakers can strategically formulate macro-level policies to control the floating range of TCEI according to the stage of development of urbanization. Doing so can achieve new urbanization, energy conservation, and emission reduction development objectives.

Considering the significantly positive influence of urban green spaces on carbon emission efficiency and the large negative effect of population respiration on carbon emissions, it should be a priority for each city's emission reduction strategy to uphold the function of carbon sinks [53]. By optimizing land use structures and enhancing the carbon capture capabilities of urban green spaces, governments and authorities can foster high-quality socio-economic development [54]. In addition, highly developed economic regions should combine local economic and social development and resource endowments to scientifically and reasonably set orderly peak targets, strictly control new carbon source land use, and make use of the carbon reduction function of potential carbon sink land use like green space [55]. They should aim to reduce the peak and concentration of carbon emissions by enhancing the degree of intensive and economical land use, resulting in a decrease in a region's total carbon emissions.

## Supporting information

**S1 Appendix. Descriptions of input and output variables.**
(PDF)

**S1 Raw data.**
(XLSX)

## Author Contributions

**Conceptualization:** Xiao-yan Liu.

**Data curation:** Fang-yi Sun.

**Formal analysis:** Qi Dai.

**Investigation:** Fang-rong Ren.

**Methodology:** Xiao-yan Liu.

**Project administration:** Fang-rong Ren.

**Software:** Fang-rong Ren.

**Visualization:** Fang-yi Sun.

**Writing – original draft:** Qi Dai.

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
