## [Decision Letter · Decision Letter 0]

29 Jul 2024

PONE-D-24-25214Ensemble intelligence predictions algorithms and land use scenarios to measure carbon emission of Yangtze River Delta region：A machine learning model based on Long Short-Term MemoryPLOS ONE

Dear Dr. Ren,

Thank you for submitting your manuscript to PLOS ONE. After careful consideration, we feel that it has merit but does not fully meet PLOS ONE’s publication criteria as it currently stands. Therefore, we invite you to submit a revised version of the manuscript that addresses the points raised during the review process.

We look forward to receiving your revised manuscript.

Kind regards,

Salim Heddam

Academic Editor

PLOS ONE

Journal Requirements:

4. We note that Figure 3 in your submission contain [map/satellite] images which may be copyrighted. All PLOS content is published under the Creative Commons Attribution License (CC BY 4.0), which means that the manuscript, images, and Supporting Information files will be freely available online, and any third party is permitted to access, download, copy, distribute, and use these materials in any way, even commercially, with proper attribution. For these reasons, we cannot publish previously copyrighted maps or satellite images created using proprietary data, such as Google software (Google Maps, Street View, and Earth). For more information, see our copyright guidelines: http://journals.plos.org/plosone/s/licenses-and-copyright.

a. You may seek permission from the original copyright holder of Figure 3 to publish the content specifically under the CC BY 4.0 license.  

Additional Editor Comments:

Reviewer 1#:

Advantages:

1.The literature review is comprehensive and logical, very logical.

2. The content is comprehensive and in-depth depth. Paper on land use of carbon efficiency related concept of the clear definition, the factors of carbon efficiency comprehensive comb, and through the empirical analysis discusses the YRD city cluster in different stages of carbon emission efficiency, including the overall efficiency, stage efficiency, important input and output index analysis and influence factors analysis, etc., detailed and in-depth content.

Disadvantages:

1.There are too many old references, and 3-5 years should account for 55% -60%. It is recommended to increase the literature quoting authoritative journal papers for 3-5 years.

2. In the analysis of the results, the explanation of some phenomena and conclusions is not deep enough, and the discussion of the mechanism behind them is lacking. It is suggested that the reasons for the difference in carbon emission reduction efficiency in different regions should be analyzed from economic structure, industrial policy, energy structure and other aspects, and more targeted suggestions and measures should be put forward.

3. The writing and expression of the paper is basically clear, but some sentences are not accurate or fluent enough, which need to be further modified and improved.

4. The formula serial number format is not uniform.

Reviewer 2#:

In the manuscript titled “Ensemble intelligence predictions algorithms and land use scenarios to measure carbon emission of Yangtze River Delta region：A machine learning model based on Long Short-Term Memory” , authors performed two-stage dynamic data envelopment analysis and LSTM models and demonstrated regional heterogeneity of carbon emission efficiency. The article is an interesting analysis and prediction of carbon emission efficiency in Yangtze River Delta urban agglomeration. This study contains some interesting findings and are valuable for the understanding of carbon reduction in urban agglomeration. However, lack of updated literature and scholarly language are the major flaw of the study. Therefore, MAJOR revision has to be done before this manuscript could be accepted for publication in the PLOS ONE.

1. The Abstract should explain the reasons for selecting cities in the Yangtze River Delta region as research subjects.

2. Abbreviation should be presented in a table. In addition, please explain their meanings when abbreviations first appear in the main text.

3. It is recommended to add relevant literature from the past three years.

4. The research gap should be listed in sections after the Literature Review.

5. The Policy Implications are too general, and the feasibility and operability of the policy implications should be improved.

6. In the Introduction section, this article adopts the form of a question to explain the research purpose but fails to clearly explain the contribution of this article.

7. In the Research Method section, it was noted that the author did not italicize some variables. Please correct this.

8. Method limitations should be briefly covered in the Research Method section

9. There are still some formatting issues in this article, please ask the author to correct them. For example, “Table 2. Changes in the efficiency of energy consumption stages by provinces in the YRD”

10. In the Descriptive Statistics section, it is evident that the data in Figures 4 and 5 should not be presented as bar charts, as the minimum values are almost imperceptible. Suggest using a box diagram instead.

11. Please do not use such kind of terms, which mean nothing in terms of scientific writing. There are other terms to be corrected too. For example, “we”.

12. There was no mention of the limitation of the study, please add the section.

13. The Discussion part is needed, but it should be placed in the result analysis section.

Reviewer 3#:

The manuscript entitled Ensemble intelligence predictions algorithms and land use scenarios to measure carbon emission of Yangtze River Delta region：A machine learning model based on Long Short-Term Memory has been evaluated. The study is meaningful, however, the draft needs Major Revisions, please see my comments:

1. Too long a title, and avoid using abbreviations in Abstract.

2. For too long Abstract, authors need to succinctly describe the research work and report the main results.

3. The research motivation is not clear, the author should gradually advance to the specific field of research from the perspective of global sustainable development.

4. The literature review is inadequate, and emerging research on land use carbon emissions needs to be considered. I will provide representative papers for your reference: https://doi.org/10.1016/j.energy.2024.131722；https://doi.org/10.1016/j.apenergy.2024.122819；https://doi.org/10.1016/j.jclepro.2023.140069；https://doi.org/10.1016/j.apenergy.2023.121488

5. Variables need to be in italics. In addition, the transfer of variables in the numerical model needs to be demonstrated with a clear system diagram.

6. What are the variables of all the equations respectively, the author needs to explain. Note that each variable corresponds to the data source studied in this paper is the focus, rather than just describing the general name of the variable. For example, t corresponds to time, and the time series is 20XX-20XX in this paper.

7. What are the sources of land use data? To my knowledge, the vast majority of land use datasets fail to identify residential and industrial land from built-up areas.

8. Descriptive statistics are not enough. In addition, LSTM, a deep learning algorithm, requires a large sample size. Has the author conducted an effect size test? In addition, I worry about the emergence of overfitting.

9. Calculating specific numbers without explaining why would call into question the contribution and significance of the manuscript.

Reviewers' comments:

Reviewer's Responses to Questions

**Comments to the Author**

1. Is the manuscript technically sound, and do the data support the conclusions?

Reviewer #1: Yes

Reviewer #2: Yes

Reviewer #3: Yes

2. Has the statistical analysis been performed appropriately and rigorously? 

Reviewer #1: Yes

Reviewer #2: Yes

Reviewer #3: Yes

3. Have the authors made all data underlying the findings in their manuscript fully available?

Reviewer #1: Yes

Reviewer #2: Yes

Reviewer #3: Yes

4. Is the manuscript presented in an intelligible fashion and written in standard English?

Reviewer #1: Yes

Reviewer #2: Yes

Reviewer #3: Yes

5. Review Comments to the Author

Reviewer #1: Advantages:

1.The literature review is comprehensive and logical, very logical.

2. The content is comprehensive and in-depth depth. Paper on land use of carbon efficiency related concept of the clear definition, the factors of carbon efficiency comprehensive comb, and through the empirical analysis discusses the YRD city cluster in different stages of carbon emission efficiency, including the overall efficiency, stage efficiency, important input and output index analysis and influence factors analysis, etc., detailed and in-depth content.

Disadvantages:

1.There are too many old references, and 3-5 years should account for 55% -60%. It is recommended to increase the literature quoting authoritative journal papers for 3-5 years.

2. In the analysis of the results, the explanation of some phenomena and conclusions is not deep enough, and the discussion of the mechanism behind them is lacking. It is suggested that the reasons for the difference in carbon emission reduction efficiency in different regions should be analyzed from economic structure, industrial policy, energy structure and other aspects, and more targeted suggestions and measures should be put forward.

3. The writing and expression of the paper is basically clear, but some sentences are not accurate or fluent enough, which need to be further modified and improved.

4. The formula serial number format is not uniform.

Reviewer #2: In the manuscript titled “Ensemble intelligence predictions algorithms and land use scenarios to measure carbon emission of Yangtze River Delta region：A machine learning model based on Long Short-Term Memory” , authors performed two-stage dynamic data envelopment analysis and LSTM models and demonstrated regional heterogeneity of carbon emission efficiency. The article is an interesting analysis and prediction of carbon emission efficiency in Yangtze River Delta urban agglomeration. This study contains some interesting findings and are valuable for the understanding of carbon reduction in urban agglomeration. However, lack of updated literature and scholarly language are the major flaw of the study. Therefore, MAJOR revision has to be done before this manuscript could be accepted for publication in the PLOS ONE.

1. The Abstract should explain the reasons for selecting cities in the Yangtze River Delta region as research subjects.

2. Abbreviation should be presented in a table. In addition, please explain their meanings when abbreviations first appear in the main text.

3. It is recommended to add relevant literature from the past three years.

4. The research gap should be listed in sections after the Literature Review.

5. The Policy Implications are too general, and the feasibility and operability of the policy implications should be improved.

6. In the Introduction section, this article adopts the form of a question to explain the research purpose but fails to clearly explain the contribution of this article.

7. In the Research Method section, it was noted that the author did not italicize some variables. Please correct this.

8. Method limitations should be briefly covered in the Research Method section

9. There are still some formatting issues in this article, please ask the author to correct them. For example, “Table 2. Changes in the efficiency of energy consumption stages by provinces in the YRD”

10. In the Descriptive Statistics section, it is evident that the data in Figures 4 and 5 should not be presented as bar charts, as the minimum values are almost imperceptible. Suggest using a box diagram instead.

11. Please do not use such kind of terms, which mean nothing in terms of scientific writing. There are other terms to be corrected too. For example, “we”.

12. There was no mention of the limitation of the study, please add the section.

13. The Discussion part is needed, but it should be placed in the result analysis section.

Reviewer #3: The manuscript entitled Ensemble intelligence predictions algorithms and land use scenarios to measure carbon emission of Yangtze River Delta region：A machine learning model based on Long Short-Term Memory has been evaluated. The study is meaningful, however, the draft needs Major Revisions, please see my comments:

1. Too long a title, and avoid using abbreviations in Abstract.

2. For too long Abstract, authors need to succinctly describe the research work and report the main results.

3. The research motivation is not clear, the author should gradually advance to the specific field of research from the perspective of global sustainable development.

4. The literature review is inadequate, and emerging research on land use carbon emissions needs to be considered. I will provide representative papers for your reference: https://doi.org/10.1016/j.energy.2024.131722；https://doi.org/10.1016/j.apenergy.2024.122819；https://doi.org/10.1016/j.jclepro.2023.140069；https://doi.org/10.1016/j.apenergy.2023.121488

5. Variables need to be in italics. In addition, the transfer of variables in the numerical model needs to be demonstrated with a clear system diagram.

6. What are the variables of all the equations respectively, the author needs to explain. Note that each variable corresponds to the data source studied in this paper is the focus, rather than just describing the general name of the variable. For example, t corresponds to time, and the time series is 20XX-20XX in this paper.

7. What are the sources of land use data? To my knowledge, the vast majority of land use datasets fail to identify residential and industrial land from built-up areas.

8. Descriptive statistics are not enough. In addition, LSTM, a deep learning algorithm, requires a large sample size. Has the author conducted an effect size test? In addition, I worry about the emergence of overfitting.

9. Calculating specific numbers without explaining why would call into question the contribution and significance of the manuscript.

6. PLOS authors have the option to publish the peer review history of their article (what does this mean?). If published, this will include your full peer review and any attached files.

Reviewer #1: No

Reviewer #2: No

Reviewer #3: No

---

## [Author Response · Author response to Decision Letter 0]

7 Aug 2024

Reviewer 1#:

Advantages:

1.The literature review is comprehensive and logical, very logical.

2. The content is comprehensive and in-depth depth. Paper on land use of carbon efficiency related concept of the clear definition, the factors of carbon efficiency comprehensive comb, and through the empirical analysis discusses the YRD city cluster in different stages of carbon emission efficiency, including the overall efficiency, stage efficiency, important input and output index analysis and influence factors analysis, etc., detailed and in-depth content.

Dear reviewer,

Thank you for giving us the opportunity to submit a revised draft of the manuscript “Ensemble intelligence predictions algorithms and land use scenarios to measure carbon emission of Yangtze River Delta region：A machine learning model based on Long Short-Term Memory” for publication in the Journal of PLOS ONE. We appreciate the time and effort that you and the reviewers dedicated to providing feedback on our manuscript and are grateful for the insightful comments on and valuable improvements to our paper. According to your nice comments, we have revised the manuscript extensively. If there are any other modifications we could make, we would like very much to modify them and we really appreciate your help. 

Disadvantages:

1. There are too many old references, and 3-5 years should account for 55% -60%. It is recommended to increase the literature quoting authoritative journal papers for 3-5 years.

Answer: Thank you for your kind comment. We have added the literature quoting authoritative journal papers for 3-5 years. (Lines 869-1004)

[1] Luo, H.J., Zhang, Y.W., Gao, X.Y., Liu, Z.G., Song, X., Meng X.Z., Yang, X.H., Yang, X.H., 2024. Unveiling land use-carbon Nexus: Spatial matrix-enhanced neural network for predicting commercial and residential carbon emissions. Energy. Volume 305. 131722, 0360-5442.

[2] Luo, H.Z., Wang, C.L., Li, C.B., Meng, X.Z., Yang X.H., Tan, Q., 2024. Multi-scale carbon emission characterization and prediction based on land use and interpretable machine learning model: A case study of the Yangtze River Delta Region, China. Applied Energy, Volume 360. 122819, 0306-2619.

[3] Weiss MC, Adusumilli S, Jagai JS, Sargis RM., 2023. Transportation-related environmental mixtures and diabetes prevalence and control in urban/ metropolitan counties in the United States. J Endocr Soc;7(6): bvad062. 

[4] Attari MYN, Ala A, Khalilpourshiraz Z., 2022. The electric power supply chain network design and emission reduction policy: a comprehensive review. Environ Sci Pollut es;29:55541–67.

[5] Wang Q, Li L, Li R., 2023. Uncovering the impact of income inequality and population aging on carbon emission efffciency: an empirical analysis of 139 countries. Sci Total Environ. 857:159508.

[6] Liu Q, Song J, Dai T, Shi A, Xu J, Wang E., 2022. Spatio-temporal dynamic evolution of carbon emission intensity and the effectiveness of carbon emission reduction at county level based on nighttime light data. J Clean Prod. 362:132301.

[7] Gorka M, Bezyk Y, Sowka I., 2021. Assessment of GHG interactions in the vicinity of the municipal waste landfill site-case study. Energies. 14(24):8259. 

[8] Wang S, Li Y, Li F, Zheng D, Yang J, Yu E., 2023. Spatialization and driving factors of carbon budget at county level in the Yangtze River Delta of China. Environ Sci Pollut Res. 23:289178.

[9] Tan, J., Wang, R., 2021. Research on evaluation and influencing factors of regional ecological efficiency from the perspective of carbon neutrality. J. Environ. Manag. 294, 113030.

[10] Fattah MA, Morshed SR, Morshed SY., 2021. Multi-layer perceptron-Markov chain-based artificial neural network for modelling future land-specific carbon emission pattern and its influences on surface temperature. SN Appl Sci; 3(3):0359. 

[11] Kang T, Wang H, He Z, Liu Z, Ren Y, Zhao P., 2023. The effects of urban land use on energy-related CO2 emissions in China. Sci Total Environ. 870:161873.

2. In the analysis of the results, the explanation of some phenomena and conclusions is not deep enough, and the discussion of the mechanism behind them is lacking. It is suggested that the reasons for the difference in carbon emission reduction efficiency in different regions should be analyzed from economic structure, industrial policy, energy structure and other aspects, and more targeted suggestions and measures should be put forward.

Answer: Thank you for the comment. We agree with your suggestions and we have merged the paragraph with the results analysis section, as can be seen in the end of section 4. In addition, we have tried our best to summarize the reasons for the difference in carbon emission reduction efficiency in different regions should be analyzed from economic structure, industrial policy, energy structure and other aspects, and more targeted suggestions and measures should be put forward. Hope to get your approval.

For example, (Lines 711-763)

“4.4 Discussion

Taking YRD as the study area, two-stage dynamic DEA, Tapio and LSTM model are used to perform measurement and prediction of TCEI in the perspective of land use. The TCEI in each city of YRD during the study interval is fluctuating, but it is generally at a high level. Nevertheless, it is still worthwhile to consider the heterogeneity of various stages and the underlying causes.

In terms of the TCEI of land use calculation, total efficiency has obvious spatial and temporal heterogeneity. It is not unexpected that provincial capitals, metropolitan areas and coastal developed cities show higher TCEI. Economically developed areas often have strong talent accumulation and resource endowment advantages, thus promoting the innovation of production technology and land use. The cumulative effect of land resources and technological innovations in cleaner production allow these cities to reduce carbon emissions while maintaining productivity. In addition, the economically developed areas have better response effect and implementation of energy conservation and emission reduction policies. Specifically, some cities in Anhui and northern Jiangsu have larger increases, while some cities’ TCEIs rose in a fluctuating state. The carbon emission efficiency of land use in cities in southern Jiangsu is moderately high. TCEI decreases in Zhejiang and Shanghai more substantially, suggesting significant potential for improvement in land use and resource allocation in these areas. Carbon emission performance varies across regions and different urbanization stages, demonstrating significant spatial and cyclical disparities.

In terms of stages, the efficiency values of most cities in the initial stage (energy consumption stage) exceeds the second stage (sustainable land utilization stage). In the initial stage, Zhejiang has the lowest total land use carbon emission efficiency among the four regions, but in the second stage, it has the second highest total efficiency value with smaller fluctuations. These improvements are a result of China’s long-term ecological preservation initiatives aimed at enhancing its cities’ green development[43-44]. In addition，among the six major input-output indicators, Jiangsu exhibits the best performance in terms of the indicators relating to carbon source of construction land and urban environmental protection input. This indicates with the advancement of urbanization and the development of industries in the province that the environmental problems in Jiangsu do not match with the final level of economic development[45].

According to the forecast results of TCEI, there is a trend of further optimization and growth of TCEI in the next six years, which indicates that the Yangtze River Delta urban agglomeration in recent years has controlled the expansion of urban built-up areas and promoted the mixed-use development of central urban areas, which has significantly promoted the improvement of carbon emission efficiency. In other words, intensive land use planning model has a positive impact on optimizing land use structure and carbon emission reduction effect[46]. The identification results of influence factors indicate that green space carbon sequestration capacity and population agglomeration effect are the core driving factors of urban land carbon emissions. Based on this, each region should seek suitable land use methods according to its own characteristics, so that the carbon emissions of each region can reach an appropriate level.

From a research perspective, this study takes the comprehensive carbon emission efficiency of land use as the research goal, focusing on the assessment of TCEI at different stages under the macro -level perspective. Land serves as an important carrier for urban economic development, while different types of land assume different functions in regional economic expansion[47-48]. In addition, 27 cities in YRD are taken as case studies for this study. There are many other studies focusing on carbon emissions throughout Asia as well as worldwide[49-50]. Therefore, future research could include other developing countries’ provinces for analysis to show clearer characteristics to help guide planning, management, and decision-making.”

”

3. The writing and expression of the paper is basically clear, but some sentences are not accurate or fluent enough, which need to be further modified and improved.

Answer: Thanks for your suggestion. The writing and expression of the paper has been modified and improved by a native English speaker from the USA. And we hope the revised manuscript could be acceptable for you.

4. The formula serial number format is not uniform.

Answer: Thank the reviewer for continuing to point out the details. We think this is an excellent suggestion. We have reordered the formula section.

For example, (Lines 259-350)

Reviewer 2#:

In the manuscript titled “Ensemble intelligence predictions algorithms and land use scenarios to measure carbon emission of Yangtze River Delta region：A machine learning model based on Long Short-Term Memory” , authors performed two-stage dynamic data envelopment analysis and LSTM models and demonstrated regional heterogeneity of carbon emission efficiency. The article is an interesting analysis and prediction of carbon emission efficiency in Yangtze River Delta urban agglomeration. This study contains some interesting findings and are valuable for the understanding of carbon reduction in urban agglomeration. However, lack of updated literature and scholarly language are the major flaw of the study. Therefore, MAJOR revision has to be done before this manuscript could be accepted for publication in the PLOS ONE.

1. The Abstract should explain the reasons for selecting cities in the Yangtze River Delta region as research subjects. (Lines 12-34)

Answer: Thank the reviewer for continuing to point out the details. We think this is an excellent suggestion. We have added the reasons for selecting cities in the Yangtze River Delta region as research subjects in the Abstract. 

“To enrich case studies for densely populated cities and highly commercialized regions, this research evaluates the total-factor carbon emission efficiency index (TCEI) of 27 cities in China's Yangtze River Delta (YRD) urban agglomeration at different stages from 2011 to 2020 using two-stage dynamic data envelopment analysis (DEA).” 

And you can see more details in 1 Introduction section:

In the choice of research objects, the YRD city cluster is the most economically efficient and urban agglomerated city area in China, accounting for 20% of China's economic output despite occupying only 2.1% of its land area. In addition, the total permanent population of YRD in 2023 is 236.9 million people, reflecting the population agglomeration effect of the YRD region. Considering the lack of case studies specifically aimed at densely populated cities and highly commercialized areas, and the need for empirical application of demonstrative methods, this paper chooses the YRD as a case to study the path of energy conservation and emission reduction of urban agglomeration with good economic endowment in developing countries. 

2. Abbreviation should be presented in a table. In addition, please explain their meanings when abbreviations first appear in the main text.

Answer: Thank the reviewer for continuing to point out the details. We think this is an excellent suggestion. We have presented Abbreviation in a table and explained their meanings when abbreviations first appear in the main text, as you can see in the second page. (Lines 37-38)

Abbreviation: 

IPCC Intergovernmental Panel on Climate Change

TCEI Total-factor carbon emission efficiency index

YRD Yangtze River Delta

SBM Slacks-based measure

DMU Decision-making unit

LMDI Logarithmic mean divisia index

DEA Data envelopment analysis

SFA Stochastic frontier model

LSTM Long Short-Term Memory

RNN Recurrent neural network

MPI Malmquist productivity index

St. dev. Standard deviation

CNY Chinese yuan

AVE Average

MSE Mean-square error

MAE Mean absolute error

3. It is recommended to add relevant literature from the past three years.

Answer: Thank you for your kind comment. We have added the literature quoting authoritative journal papers from the past three years. (Lines 869-1004)

[1] Luo, H.J., Zhang, Y.W., Gao, X.Y., Liu, Z.G., Song, X., Meng X.Z., Yang, X.H., Yang, X.H., 2024. Unveiling land use-carbon Nexus: Spatial matrix-enhanced neural network for predicting commercial and residential carbon emissions. Energy. Volume 305. 131722, 0360-5442.

[2] Luo, H.Z., Wang, C.L., Li, C.B., Meng, X.Z., Yang X.H., Tan, Q., 2024. Multi-scale carbon emission characterization and prediction based on land use and interpretable machine learning model: A case study of the Yangtze River Delta Region, China. Applied Energy, Volume 360. 122819, 0306-2619.

[3] Weiss MC, Adusumilli S, Jagai JS, Sargis RM., 2023. Transportation-related environmental mixtures and diabetes prevalence and control in urban/ metropolitan counties in the United States. J Endocr Soc;7(6): bvad062. 

[4] Attari MYN, Ala A, Khalilpourshiraz Z., 2022. The electric power supply chain network design and emission reduction policy: a comprehensive review. Environ Sci Pollut es;29:55541–67.

[5] Wang Q, Li L, Li R., 2023. Uncovering the impact of income inequality and population aging on carbon emission efffciency: an empirical analysis of 139 countries. Sci Total Environ. 857:159508.

[6] Liu Q, Song J, Dai T, Shi A, Xu J, Wang E., 2022. Spatio-temporal dynamic evolution of carbon emission intensity and the effectiveness of carbon emission reduction at county level based on nighttime light data. J Clean Prod. 362:132301.

[7] Gorka M, Bezyk Y, Sowka I., 2021. Assessment of GHG interactions in the vicinity of the municipal waste landfill site-case study. Energies. 14(24):8259. 

[8] Wang S, Li Y, Li F, Zheng D, Yang J, Yu E., 2023. Spatialization and driving factors of carbon budget at county level in the Yangtze River Delta of China. Environ Sci Pollut Res. 23:289178.

[9] Tan, J., Wang, R., 2021. Research on evaluation and influencing factors of regional ecological efficiency from the perspective of carbon neutrality. J. Environ. Manag. 294, 113030.

[10] Fattah MA, Morshed SR, Morshed SY., 2021. Multi-layer perceptron-Markov chain-based artificial neural network for modelling future land-specific carbon emission pattern and its influences on surface temperature. SN Appl Sci; 3(3):0359. 

[11] Kang T, Wang H, He Z, Liu Z, Ren Y, Zhao P., 2023. The effects of urban land use on energy-related CO2 emissions in China. Sci Total Environ. 870:161873.

4. The research gap should be listed in sections after the Literature Review.

Answer：Thank you for your insightful comment and kind suggestion. We agree with the reviewer’s assessment. Therefore, we have tried our best to summarize the research gap in the end of the Literature Review.

(Lines 207-223)

“In summary, some shortcomings are also found from other studies' results. First, the majority of the studies have been focused on the accounting for TCEI of specific land use types, thereby overlooking macro-level land use efficiency measurement. This also results in a decline in the number of analyses of the driving factor

---

## [Decision Letter · Decision Letter 1]

15 Aug 2024

PONE-D-24-25214R1Ensemble intelligence predictions algorithms and land use scenarios to measure carbon emission of Yangtze River Delta region：A machine learning model based on Long Short-Term MemoryPLOS ONE

Dear Dr. Ren,

Thank you for submitting your manuscript to PLOS ONE. After careful consideration, we feel that it has merit but does not fully meet PLOS ONE’s publication criteria as it currently stands. Therefore, we invite you to submit a revised version of the manuscript that addresses the points raised during the review process.

We look forward to receiving your revised manuscript.

Kind regards,

Salim Heddam

Academic Editor

PLOS ONE

Additional Editor Comments:

Reviewer 1#:Specific Comments.pdf

Reviewer 2#:The comments previously made have been greatly improved. I believe that the current level of this manuscript is acceptable for publication.

Reviewer 3#:The revisions are well addressed. The revisions can be accepted as is. No further review comments for revisions are needed. The reviewer has studied the revisions made by the authors and found that all the comments and suggestions have been well addressed by the authors.

Reviewers' comments:

Reviewer's Responses to Questions

**Comments to the Author**

1. If the authors have adequately addressed your comments raised in a previous round of review and you feel that this manuscript is now acceptable for publication, you may indicate that here to bypass the “Comments to the Author” section, enter your conflict of interest statement in the “Confidential to Editor” section, and submit your "Accept" recommendation.

Reviewer #1: (No Response)

Reviewer #2: All comments have been addressed

Reviewer #3: All comments have been addressed

2. Is the manuscript technically sound, and do the data support the conclusions?

Reviewer #1: (No Response)

Reviewer #2: Yes

Reviewer #3: Yes

3. Has the statistical analysis been performed appropriately and rigorously? 

Reviewer #1: (No Response)

Reviewer #2: Yes

Reviewer #3: Yes

4. Have the authors made all data underlying the findings in their manuscript fully available?

Reviewer #1: (No Response)

Reviewer #2: Yes

Reviewer #3: Yes

5. Is the manuscript presented in an intelligible fashion and written in standard English?

Reviewer #1: (No Response)

Reviewer #2: Yes

Reviewer #3: Yes

6. Review Comments to the Author

Reviewer #1: (No Response)

Reviewer #2: The comments previously made have been greatly improved. I believe that the current level of this manuscript is acceptable for publication.

Reviewer #3: The revisions are well addressed. The revisions can be accepted as is. No further review comments for revisions are needed. The reviewer has studied the revisions made by the authors and found that all the comments and suggestions have been well addressed by the authors.

7. PLOS authors have the option to publish the peer review history of their article (what does this mean?). If published, this will include your full peer review and any attached files.

Reviewer #1: No

Reviewer #2: No

Reviewer #3: No

---

## [Author Response · Author response to Decision Letter 1]

2 Sep 2024

Dear reviewer,

Thank you for giving us the opportunity to submit a revised draft of the manuscript “Ensemble intelligence prediction algorithms and land use scenarios to measure carbon emissions of the Yangtze River Delta: A machine learning model based on Long Short-Term Memory” for publication in the Journal of PLOS ONE. We appreciate the time and effort that you and the reviewers dedicated to providing feedback on our manuscript and are grateful for the insightful comments on and valuable improvements to our paper. We have incorporated most of the suggestions made by the reviewers. In the revised manuscript, all the changes are highlighted in yellow for easy inspection. Please see below, for a point-by-point response to the reviewers’ comments and concerns. According to the reviewer’s comments, we have revised the manuscript extensively. 

Disadvantages: 

1.Insufficient interpretation of the results: in the analysis of the results, the interpretation of some phenomena and conclusions is not deep enough, and there is a lack of discussion on the mechanism behind them. For example, the analysis of the causes for the differences in carbon emission efficiency in different regions is not comprehensive enough, so more in-depth analysis should be conducted from the aspects of economic structure, industrial policy, energy structure and so on, and more targeted suggestions and measures should be put forward.

Answer: Thank you for the comment. We agree with your suggestions and we have merged the paragraph with the results analysis section, as can be seen in the section 3, 4 and 5. In addition, we have tried our best to summarize the reasons for the difference in carbon emission reduction efficiency in different regions and analyzed from economic structure, industrial policy, energy structure and other aspects. Hope to get your approval. In addition, more targeted suggestions and measures have been put forward.

For example, 

Anhui and northern Jiangsu show a larger increase, especially Anqing, which rises from 0.5308 at the beginning to 0.8872 later. This may be because these latecomer cities, in the stage of new urbanization, timely changed their original extensive land use patterns, abandoned the original concept of promoting economic development with heavy industry as the pillar industry, and carried out corresponding industrial transformation, accelerating the construction of green spaces in new urban areas, thereby reducing carbon emissions under the same energy consumption. (Lines 435-442)

This may be due to the rapid urban expansion stage, where the government converted more and more urban green space into construction land to promote economic development, thus greatly reducing the carbon sequestration capacity of green space. (Lines 451-467)

In the second stage (Figure 7), Anhui Province performed the best in terms of efficiency in the sustainable land use stage from 2011 to 2020, maintaining a stable average of 0.7127 and continuously improving every year. The reason is that due to a certain gap in the total GDP of Anhui compared to Jiangsu, Zhejiang, and Shanghai, Anhui's carbon emissions level is lower than that of Jiangsu, Zhejiang, and Shanghai. The financial investment related to environmental protection is similar among different cities, and the reduction of energy consumption waste and garbage caused by population outflow has led Anhui to invest more in urban green space construction. (Lines 499-533)

In terms of the TCEI of land use calculation, total efficiency has obvious spatial and temporal heterogeneity. It is not unexpected that provincial capitals, metropolitan areas, and coastal developed cities show higher TCEI. This is because as the provincial capitals of their respective provinces, although these cities have a relatively high population size and built-up area, they have already relocated heavy polluting industries and carried out a large amount of urban green space construction in conjunction with the existing suburban green spaces in the city during the urban development process. (Lines 724-753)

2. Writing expression needs to be improved: some sentences in the paper are not accurate or fluent, and need to be further modified and improved. For example, avoid using unprofessional expressions such as "we". It is suggested to improve the writing quality and expression accuracy of the paper.

Answer: Thanks for your suggestion. The writing and expression of the paper has been modified and improved by a native English speaker from the USA. And we hope the revised manuscript could be acceptable for you. Moreover, we have tried to change “We” to “This research”, " Meanwhile " to “In addition” or “Moreover” in order to be more professional, as seen in Abstract section. The certification in English editing are as follows.

For example, 

5.1 Conclusions

This research employs a dynamic two-stage DEA model to build a TCEI assessment framework. Subsequently, the study determines TCEI in the land use process of YRD from 2011 to 2020 and further predict their dynamic evolution. The findings suggest the following. (Lines 785-795)

3. Insufficient interpretation of variables: the data source of each variable is not clearly specified in the paper; the large sample problems required by the deep learning algorithm LSTM were not fully considered, the effect size test was not fully conducted, and the concerns about possible overfitting problems were not effectively solved. It is suggested that the variable meaning and data sources of all equations be explained in detail in the paper to enhance the comprehensibility of the paper. 

Answer: Thank the reviewer for continuing to point out the details. We think this is an excellent suggestion. We first supplement the effect size test, explain the model overfitting problem, and add explanatory notes on the sources of variables.

1. Supplement on effect size test

To avoid some problems like the size of the effect is not practical or of little value, Cohen (1992) suggested fixing the power at 0.80 (β = 0.20), which is also a convention proposed for general use. The approach of Dupont and Plummer (1990) is used for paired and independent samples. The ratio of number of patients in the samples being compared may be specified by the user. This method produces results that are in close agreement with those of Pearson and Hartley (1970).

Based on this, this research is planning a study with 44 experimental subjects and 176 control subjects. In addition, the Type I error probability associated with this test of this null hypothesis (α) is 0.1. In a previous study the response within each subject group was normally distributed with standard deviation 0.168. If the true difference in the experimental and control means is 0.071, we will be able to reject the null hypothesis that the population means of the experimental and control groups are equal with probability (power) 0.804. This suggests that our experimental and control data are reasonable. The following is a statistical power graph based on the calculation results.

References:

http://dx.doi.org/10.20982/tqmp.14.4.p242.

https://psycnet.apa.org/doi/10.1037/a0024338.

https://psycnet.apa.org/doi/10.1037/0021-9010.90.1.94.

2. Testing and optimization of overfitting problems

First of all, the fitting results of our four variables are good, and the adjusted R2 is at the level of 0.9. Secondly, we use the learning curve to determine whether there is overfitting phenomenon in the prediction of efficiency value. As shown in the figure, y (efficiency) training set and verification set both have lower loss errors. On the one hand, the training set loss gradually decreases and flattens with the increase of training samples, indicating that adding more training samples does not improve the performance of the model on training data. On the other hand, the learning curve of the training set has a high validation loss at the beginning, and gradually decreases and tends to be flat with the increase of training samples. Therefore, it is judged that there is no overfitting phenomenon in the prediction of the efficiency value y(efficiency).

Figure 1. Plot of Input Variable Fit.

Figure 2. Graph of training set and test set learning curve

In addition, in view of the possibility of slight fitting in small samples, this research adopted regularization and random silencing of some neurons during each training to reduce the complexity of the model and the degree of dependence between neurons, so as to enhance the generalization ability of the model.

References:

Ying, X. (2019, February). An overview of overfitting and its solutions. In Journal of physics: Conference series (Vol. 1168, p. 022022). IOP Publishing.

Hawkins, D. M. (2004). The problem of overfitting. Journal of chemical information and computer sciences, 44(1), 1-12.

Vul, E., Goodman, N., Griffiths, T. L., & Tenenbaum, J. B. (2014). One and done? Optimal decisions from very few samples. Cognitive science, 38(4), 599-637.

3. Supplement on the sources of variables

We have supplemented the data sources and the meaning of variables in the table, as can be seen in Appendix A. (Descriptions of input and output variables)

(Line 1028-1029)

---

## [Decision Letter · Decision Letter 2]

19 Sep 2024

Ensemble intelligence prediction algorithms and land use scenarios to measure carbon emissions of the Yangtze River Delta: A machine learning model based on Long Short-Term Memory

PONE-D-24-25214R2

Dear Dr.  Ren

We’re pleased to inform you that your manuscript has been judged scientifically suitable for publication and will be formally accepted for publication once it meets all outstanding technical requirements.

Kind regards,

Salim Heddam

Academic Editor

PLOS ONE

Reviewers' comments:

Reviewer's Responses to Questions

**Comments to the Author**

1. If the authors have adequately addressed your comments raised in a previous round of review and you feel that this manuscript is now acceptable for publication, you may indicate that here to bypass the “Comments to the Author” section, enter your conflict of interest statement in the “Confidential to Editor” section, and submit your "Accept" recommendation.

Reviewer #1: (No Response)

2. Is the manuscript technically sound, and do the data support the conclusions?

Reviewer #1: (No Response)

3. Has the statistical analysis been performed appropriately and rigorously? 

Reviewer #1: (No Response)

4. Have the authors made all data underlying the findings in their manuscript fully available?

Reviewer #1: (No Response)

5. Is the manuscript presented in an intelligible fashion and written in standard English?

Reviewer #1: (No Response)

6. Review Comments to the Author

Reviewer #1: (No Response)

7. PLOS authors have the option to publish the peer review history of their article (what does this mean?). If published, this will include your full peer review and any attached files.

Reviewer #1: No

---

## [Editor Report · Acceptance letter]

25 Sep 2024

PONE-D-24-25214R2 

PLOS ONE

Dear Dr. Ren, 

I'm pleased to inform you that your manuscript has been deemed suitable for publication in PLOS ONE. Congratulations! Your manuscript is now being handed over to our production team.

Kind regards, 

on behalf of

Dr. Salim Heddam 

Academic Editor

PLOS ONE